# Proximal Regularization of Deep Residual Neural Networks Applied to High-dimensional Genomic Data

## Abstract

Residual neural networks (ResNets) have become widely used as they allow for smooth and efficient training of deep neural network architectures. However, when trained on small, noisy and high-dimensional data, ResNets may suffer from overfitting due to the large amount of parameters. As a solution, a range of regularization methods have been proposed. One promising approach relies on the proximal mapping technique which is computationally efficient since it can be directly incorporated into the optimization algorithm. However, the performance of ResNets with various convex or non-convex proximal regularizers remains under-explored on high-dimensional data. In this study, we propose an extended stochastic adaptive proximal gradient ResNet method that can handle both convex and non-convex regularizers that range from $L_0$ to $L_\infty$. Moreover, we evaluate the prediction performance in a supervised regression setting on four real high-dimensional genomic datasets from mice, pig, wheat and loblolly pine. For comparison, we also implement and evaluate traditional sparse linear proximal methods with the same regularizers, as well as LightGBM. Experimental results demonstrate that an 18-layer ResNet with $L_{\frac{1}{2}}$ regularization outperforms other configurations on both mice and pig datasets. For the wheat and loblolly pine data, the 15-layer ResNet $L_{\frac{1}{2}}$ configuration achieves the lowest test mean squared errors. These findings highlight the effectiveness of the regularized adaptive proximal gradient ResNet method and its potential for prediction tasks on high-dimensional genomic data.

## 1 Introduction

ResNets were introduced to address the difficulties associated with training of very deep neural networks (He et al., 2016a). The main innovation of ResNets are skip connections which enable the network to learn residual functions with reference to the layer inputs instead of learning unreferenced functions. This architecture is beneficial in addressing the vanishing and exploding gradient problems. ResNets have emerged as a powerful tool for nonlinear regression and classification, demonstrating robustness to extrapolation and handling "out-of-distribution" data samples effectively (Bondarenko, 2021; Ge et al., 2022). By incorporating shortcut residual connections, it has become possible to train much deeper multilayer perceptrons (MLPs) and convolutional neural networks (CNNs) than was previously possible. ResNets have achieved record low error rates, some important examples include Wide ResNet (Zagoruyko, 2016), ResNeXt (Xie et al., 2017) and PyramidNet (Han et al., 2017). However, the ResNet architecture is on its own not sufficient to reduce the generalization error because of overfitting.

To improve the performance of ResNets, different regularization methods that place more coherent constraints on the model parameters have been proposed (Miyato et al., 2018; Santos & Papa, 2022). Data augmentation techniques artificially increase the diversity of training data, helping to increase the robustness of the model (Shorten & Khoshgoftaar, 2019). Early stopping halts training when the validation loss ceases to improve, preventing the model from overfitting (Prechelt, 2012). Label smoothing softens hard labels, preventing overconfidence in predictions and improving generalization (Müller et al., 2019). The dropout randomly deactivates neurons during training, forcing networks to learn redundant features, which reduces overfitting (Srivastava et al., 2014). Other widely used regularization methods include stochastic

gradient descent (Zhang et al., 2021) and weight decay (Krogh & Hertz, 1991). Batch normalization techniques allow for smooth gradient flow and stable training (Ioffe, 2015; Santurkar et al., 2018). Unfortunately, the regularization effect of these approaches is generally too weak on high-dimensional input data.

High-dimensional data analysis is a rapidly growing field of research that focuses on data with a large number of input variables, often much larger than the sample size. For example, high-throughput measurements in genomics often contain thousands or millions of variables, such as single nucleotide polymorphism (SNP) markers and gene expression data, for each individual observation (Davey et al., 2011; de Los Campos et al., 2013). When applied to this domain, deep learning models face two pronounced challenges: the inherent noise within the data from biological and environmental factors, and the presence of complex, non-linear interactions between markers, such as linkage disequilibrium and epistasis, which are difficult for traditional models to capture (Montesinos-Lopez et al., 2024; Okut et al., 2011). To effectively model these intricate relationships, a deep architecture is necessary. However, traditional neural networks often suffer from vanishing gradients in deep configurations, making training intractable. ResNets, with their residual connections, offer a solution by enabling the stable training of very deep networks capable of modeling these complex genomic interactions. While deep models are powerful, their high number of parameters can easily lead to severe overfitting on noisy genomic data, a problem that standard regularization methods often fail to address adequately.

In this study, our main contributions can be summarized as follows:

1. We present an extended stochastic adaptive proximal gradient framework for deep ResNets. Our approach incorporates a broad range of regularization functions: including both convex and non-convex penalties from $L_0$ to $L_\infty$ to effectively enforce sparsity and control model complexity in high-dimensional settings. We also give details of its convergence to show its theoretical rationality.

2. We provide a systematic and in-depth evaluation of this proposed method on four diverse, high-dimensional genomic datasets from mice, pig, wheat, and loblolly pine. This comprehensive analysis evaluates the impact of various regularization functions and network sizes on predictive accuracy, providing valuable insights into optimal model configurations for this challenging domain.

3. We compare the performance of our regularized ResNet method against both traditional sparse linear proximal methods and the popular machine learning algorithm LightGBM. This comparison highlights the superior predictive performance of our deep learning approach on typical genomic datasets.

Unlike prior work that has focused on a limited set of regularizers (Yun et al., 2021), our framework incorporates a broad range of regularization functions that can enforce specific sparsity constraints, which are particularly beneficial for high-dimensional genomic data (Fan et al., 2024). Adaptive proximal gradient methods allow dynamic adjustments to learning rates while simultaneously incorporating regularization terms, effectively balancing model complexity, weight sparsity and predictive accuracy. This dual capability makes them particularly well-suited for high-dimensional datasets, where many input variables are redundant.

## 2 Related Work

### 2.1 Regularization in Deep Neural Networks

To improve the training and performance of a deep neural network, the issues of shallow versus deep networks have been extensively discussed in machine learning (Larochelle et al., 2007). The ResNet architecture, introduced by He et al. (2016a), addresses the depth problem by allowing the construction of very deep architectures without suffering from vanishing or exploding gradient issues, which typically limits gradient flow during backpropagation. The core innovation is the introduction of "skip connections" that enable the

network to learn residual functions, where the layer output is a combination of the input and the learned residual. This identity mapping has been shown to accelerate training in very deep networks, as it allows gradients to pass through unchanged across multiple layers (He et al., 2016b). Building on this, Wide ResNets (Zagoruyko, 2016) were designed to prioritize network width while reducing depth, demonstrating that increasing width can be more effective than increasing depth for certain problems. Beyond architectural design, training-based regularization methods are crucial for enhancing generalization and robustness of ResNets. Dropout (Srivastava et al., 2014) is one of the most popular methods for reducing overfitting, working by randomly deactivating a fraction of neurons during each forward pass. This prevents co-adaptation of neurons and forces the network to learn more robust features. Early stopping (Prechelt, 2012) is a simple but powerful technique that halts training when validation error starts to increase, preventing the model from overfitting to the training data. Other effective techniques include data augmentation (Shorten & Khoshgoftaar, 2019), which enriches the training dataset by applying transformations to the input data, and Label smoothing (Müller et al., 2019), which softens hard labels to prevent the model from becoming overconfident.

Normalization techniques have also been introduced to improve gradient flow and stabilize training. Batch Normalization (BatchNorm) (Ioffe, 2015; Wu et al., 2024) has become a standard component in modern CNNs, normalizing the activations of a layer's input across a mini-batch. This helps to combat the internal covariate shift problem and improves both the speed and stability of training. In contrast, layer normalization (LayerNorm) (Ba, 2016) and weight normalization (WeightNorm) (Salimans & Kingma, 2016) are alternative methods that are often better suited for recurrent neural networks (RNNs) and MLPs, respectively, as they are not dependent on batch size.

## 2.2 Adaptive Optimization Algorithms

While the regularization methods discussed above modify the network architecture or training data, optimization algorithms directly control how model parameters are updated. Adaptive gradient methods are an important class of optimizers that adjust learning rates during training, which is particularly useful when fixed learning rate schedules underperform (Zhao & Huang, 2023). The traditional stochastic gradient descent (SGD) method, while simple and effective for many problems, can suffer from slow convergence and is highly sensitive to the initial learning rate selection. Even to increase the performance slightly would require significantly more parameters due to the extra layers or increasing the number of filters per layer (Bui et al., 2021).

Adaptive methods address these limitations by maintaining a dynamic learning rate for each parameter. Adagrad (Duchi et al., 2011), one of the earliest adaptive methods, modifies the learning rate for each parameter based on the historical sum of squared gradients. This approach gives larger updates to infrequent features and smaller updates to frequent ones. However, a major drawback is that the learning rate tends to shrink excessively over time, limiting its effectiveness for long-term training. RMSProp (Tieleman & Hinton, 2012) was developed to address this by maintaining a moving average of squared gradients, which prevents the learning rates from decaying too quickly. The Adam optimizer (Kingma, 2014) further builds on RMSProp by incorporating momentum, maintaining two moving averages: one for the gradient (first moment) and one for the squared gradient (second moment). This combination of momentum and adaptive learning rates makes Adam one of the most widely used and effective optimizers in deep learning (Soydaner, 2020).

## 2.3 Proximal Regularization and Optimization

Proximal regularization refers to a powerful family of optimization methods that combines a traditional gradient descent step with a proximal mapping to handle non-smooth penalties. This approach allows for imposing structural constraints, such as sparsity, on model parameters that are not possible with standard optimization algorithms like SGD or Adam alone. A key advantage is the ability to decouple the optimization

of a smooth loss function from the optimization of a non-smooth regularizer, allowing both to be handled efficiently.

Proximal gradient methods are a cornerstone of this family, incorporating a regularization step that allows for non-smooth constraints to be imposed on the model parameters (Parikh & Boyd, 2014; Beck, 2017; Bui et al., 2021). Proximal methods have also been integrated with adaptive first-order methods like Adam and RMSProp, enhancing their ability to handle non-smooth regularizers. For instance, AdaProx (Melchior et al., 2019) uses an adaptive step size scheme to avoid the need for explicit computation of the Lipschitz constants, reducing per-iteration cost. Furthermore, weighted proximal operators have been proposed to incorporate the adaptive learning rates of Adam and RMSProp (Ukil et al., 2021). The ProxGen framework (Yun et al., 2021) provides a unified approach that combines adaptive optimization with proximal regularization, enabling the use of a wide range of non-convex and non-differentiable penalties within deep learning. More recently, nonconvex stochastic Bregman proximal gradient methods have been developed to provide convergence guarantees and practical improvements for deep learning applications (Ding et al., 2025).

In applied genomics, deep networks with nonconvex penalties have started to emerge. For example, MCP-regularized deep neural networks (DNN-MCP) have been applied in genomic selection studies and demonstrated improved predictive performance compared to linear baselines (Lv et al., 2022). Comparative analyses of concave penalties (MCP, SCAD) and fractional penalties such as $L_{1/2}$ further highlight that fractional penalties often provide a better trade-off between sparsity and accuracy, while MCP and SCAD may lead to instability in deeper networks (Luo & Halabi, 2023). Reviews of genomic prediction methods also emphasize the growing role of deep models, particularly when combined with advanced regularization strategies (Montesinos-López et al., 2024; Lourenço et al., 2024).

## 3    Methods

### 3.1    Problem Formulation

ResNets can be susceptible to overfitting when trained on small, noisy or high-dimensional datasets due to their deep architecture and large number of parameters (Hinton et al., 2012). Regularization is an essential strategy to mitigate this issue by imposing constraints on the network's structure and enhancing generalization (Ukil et al., 2021). The optimization problem for training networks with regularization can be formalized as:

$$\underset{\theta \in \mathbb{R}^d}{\text{minimize}} \quad \mathcal{F}(\theta) = \mathcal{L}(\theta) + \lambda \mathcal{R}(\theta), \tag{1}$$

where $\theta \in \mathbb{R}^d$ denotes the network parameters, $\mathcal{L}(\theta)$ is the empirical loss function, $\lambda > 0$ is a regularization parameter and $\mathcal{R}(\theta)$ is a data-independent regularizer, which encourages more effective models and leads to a better estimation (Bousquet & Elisseeff, 2002). Proximal gradient methods are particularly well-suited for solving such optimization problems as they incorporate regularization terms, including non-differentiable ones, directly into the training process (Liang et al., 2023). These methods can ensure that the network parameters are within reasonable bounds and prevent them from becoming excessively large, which would otherwise have a negative impact on predicting unseen data.

Minimizing the above objective function defined is typically accomplished using stochastic first-order optimization techniques applied to the $k$-th random mini-batch of the data:

$$\underset{\theta \in \mathbb{R}^d}{\text{minimize}} \quad \mathcal{F}(\theta) = \sum_{k=0}^{n-1} \mathcal{F}_k(\theta). \tag{2}$$

### 3.2    Adaptive Proximal Gradient Descent Methods for ResNets with Various Regularizer

Adaptive proximal gradient descent methods (Duchi et al., 2011; Tieleman & Hinton, 2012) adjust the learning rate based on past gradient information, introducing a form of preconditioning by scaling gradients based on their historical magnitudes. By combining proximal updates with adaptive preconditioning, the recently introduced ProxGEN offers a robust and flexible optimization method that addresses complex

regularization of both non-smooth and non-convex regularization terms in neural network training (Yun et al., 2021). Based on the Adam optimizer (Kingma, 2014) and the idea of diagonal preconditioners (Pock & Chambolle, 2011), the preconditioner for ProxGEN can be formulated as $C_t = \sqrt{\beta_2 C_{t-1} + (1 - \beta_2)g_t^2}$, where $g_t$ is the gradient of a stochastic mini-batch at iteration $t$ and $\beta_2 \in [0, 1)$. Given a non-empty proximal mapping

$$\text{prox}_f(\theta) = \arg\min_z \{f(z) + \frac{1}{2}\|z - \theta\|^2\}, \tag{3}$$

the update rule for the weight parameters then becomes

$$\theta_{t+1} = \text{prox}_{\alpha_t \lambda \mathcal{R}(\cdot)}^{C_t + \delta I}(\theta_t - \alpha_t(C_t + \delta I)^{-1}m_t), \tag{4}$$

where $m_t = \beta_1 m_{t-1} + (1 - \beta_1)g_t$ represents the first-order momentum, $\theta_t$ is the parameter vector at iteration $t$, $\lambda > 0$ is the penalty factor, $\alpha_t$ is the adaptive learning rate and $\theta_{t+1}$ is the new parameter vector at iteration $t + 1$. The small constant $\delta$ ensures the positive definiteness of the matrix $C_t + \delta I$, which is crucial for the stability of algorithm. The parameter update at each iteration is governed by the gradient of the loss function, the regularizer, the learning rate $\alpha_t$, and the penalty factor $\lambda$. One of the key challenges with the momentum $m_t$ in above equation is the initialization bias that occurs when $m_{t-1}$ is set to zero at the start of optimization. When $g_t$ is large, this bias will hinder convergence, especially early in training. The bias correction method in Adam is designed to mitigate this initialization bias using the decay rates $\beta_1$ and $\beta_2$ for the first and second moments, respectively. Algorithm 1 outlines the full ProxGEN method for a general choice of differentiable loss and closed-form proximal regularization functions.

---

**Algorithm 1** Adaptive Stochastic Proximal Gradient Descent for ResNets with Regularization

---

**Require:** Stepsize sequence $\{\alpha_t\}_{t=1}^T$, momentum parameters $\beta_1 \in [0, 1)$, $\beta_2 \in [0, 1)$, regularization parameter $\lambda$, small constant $\delta > 0$, regularization function $\mathcal{R}$ (e.g., $L_q$, MCP, SCAD).
1: **Initialize:** Network parameters $\theta_1 \in \mathbb{R}^p$, momentum $m_0 = 0 \in \mathbb{R}^p$, preconditioner matrix $C_0 = 0 \in \mathbb{R}^{p \times p}$.
2: **for** $t = 1, 2, \ldots, T$ **do**
3:     Compute the gradient on a random mini-batch: $g_t \leftarrow \nabla_\theta \mathcal{L}(\theta_t)$
4:     Update the momentum: $m_t \leftarrow \beta_1 m_{t-1} + (1 - \beta_1)g_t$
5:     Update the preconditioner: $C_t \leftarrow \sqrt{\beta_2 C_{t-1} + (1 - \beta_2)g_t^2}$
6:     Update the weight parameters using the proximal operator:

$$\theta_{t+1} = \text{prox}_{\alpha_t \lambda \mathcal{R}}^{C_t + \delta I}\left(\theta_t - \alpha_t(C_t + \delta I)^{-1}m_t\right)$$

7: **end for**

---

We now take a look at how to generalize the regularization function $\mathcal{R}(\theta)$. The closed-form solution for a general $L_q$ regularization problem can be written as

$$\hat{\theta} = \arg\min_\theta \{(\theta - z)^2 + \lambda\|\theta\|^q\}, \tag{5}$$

where $\mathcal{R}(\theta) = \|\theta\|^q$ represents the $L_q$ penalty function for $0 \le q \le \infty$. We extend the application of sparse $L_q$ regularization beyond the conventional range $0 \le q \le 1$ to encompass cases where $q > 1$, and therefore end up with regularizers with the following penalty functions: $L_0, L_{\frac{1}{2}}, L_{\frac{2}{3}}, L_1, L_{\frac{4}{3}}, L_{\frac{3}{2}}, L_2, L_3, L_4$ and $L_\infty$. In addition, we also explore the smoothly clipped absolute deviation (SCAD) (Mehranian et al., 2013) and the minimax concave penalty (MCP) (Zhang, 2010) regularizers. This set of regularizers provides deeper insights into the behavior of various proximal operators and their effects on ResNets in the case of high-dimensional genomic datasets. Figure 1 compares these non-convex and convex penalty functions and their corresponding proximal operators for a single parameter $\theta \in \mathbb{R}$. The mathematical details of the different regularization functions and their proximal operators are provided in the Appendix A. Convergence analysis is given in Appendix B.

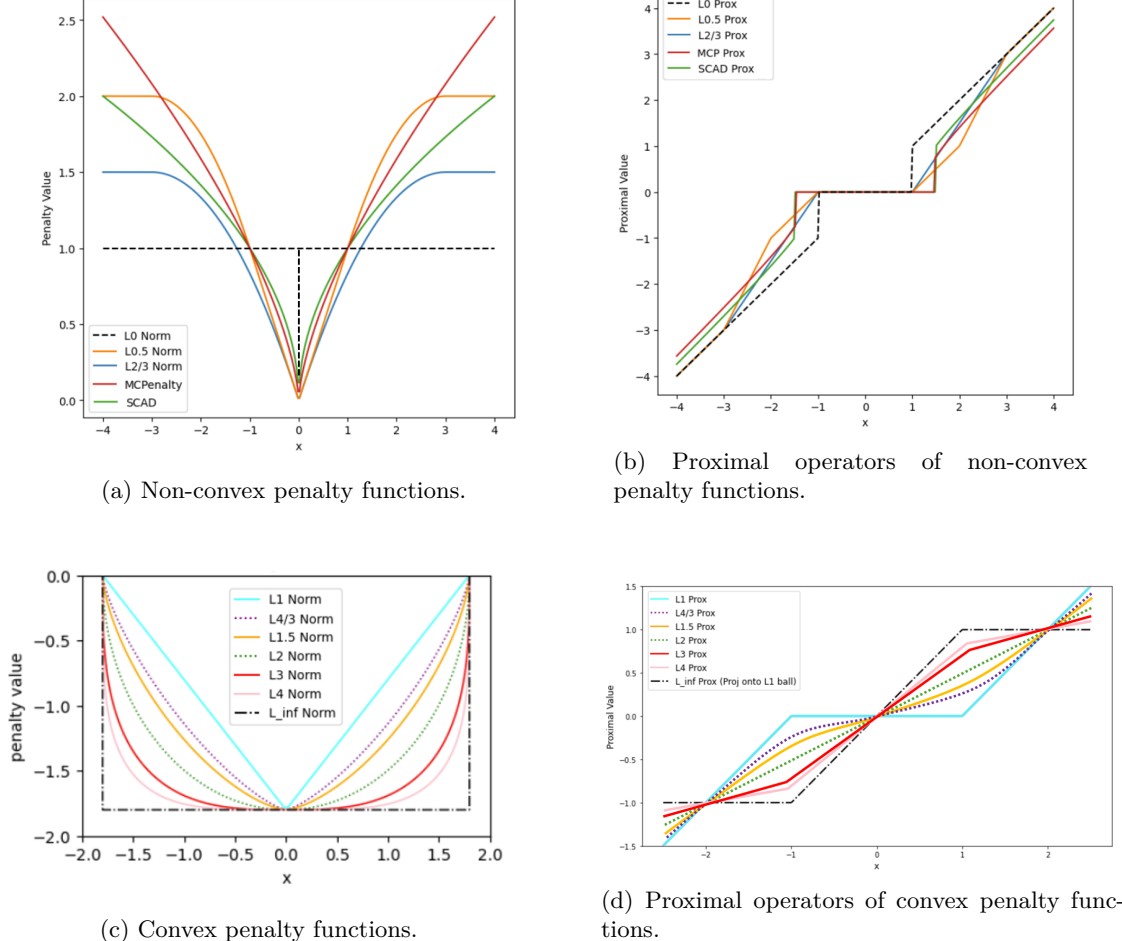

(a) Non-convex penalty functions.

(b) Proximal operators of non-convex penalty functions.

(c) Convex penalty functions.

(d) Proximal operators of convex penalty functions.

Figure 1: Comparison of non-convex and convex penalty functions and their proximal operators.

### 3.3 Network Architectures

We adopt a highly flexible design following ResNets. The network consists of basic block ResNets from 5 layers to 18 layers and of bottleneck block ResNets beyond 18 layers. Figure 2 shows the basic block and the bottleneck block used in this work. A residual block can be formulated as

$$X_{l+1} = X_l + \mathcal{E}(X_l, \theta_l), \tag{6}$$

where $X_l$ and $X_{l+1}$ are the input and the output of the $l$-th unit in the network, respectively, and $\mathcal{E}$ is a sequence of 1D convolution, batch normalization and ReLU activation, followed by another 1D convolution and batch normalization. As described in (He et al., 2016a), a ResNet consists of sequentially stacked residual blocks. For the the basic block, it contains two consecutive $1 \times 1$ convolutions with batch normalization and ReLU proceeding convolution, where a stride of 2 is used for dimension reduction in the $1 \times 1$ convolution. For the bottleneck block, ResNet utilizes a $1 \times 1$ or a $3 \times 3$ convolution to adjust the number of channels, where a stride of 1 is applied to the other convolutions to improve computational efficiency. In comparison to previous architectures (Zagoruyko, 2016), the use of varying strides and residual blocks mitigates the loss of information at each layer.

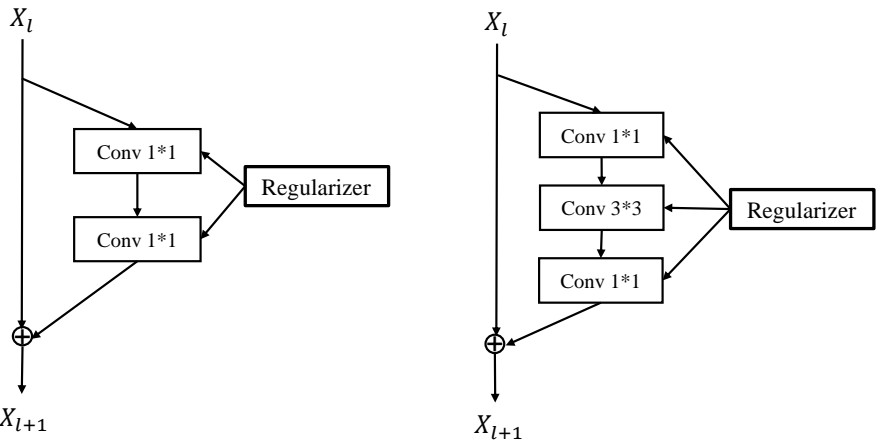

Figure 2: Residual Functions $\mathcal{E}$. Left panel: a basic block used in ResNets with 5 to 18 layers. Right panel: a bottleneck block with three convolutional layers used in ResNets with 20 to 25 layers. Batch normalization and ReLU activation precede each convolution (omitted for clarity).

## 4    Material

Mice data: The first data of this study is the mice data. It is a part of the BGLR package in R (Pérez & de Los Campos, 2014), but originally comes from the Wellcome Trust (http://gscan.well.ox.ac.uk) and has been used for whole-genome regression in several earlier studies (Legarra et al., 2008; Okut et al., 2011). It consists of genotypes and phenotypes of 1,814 mice. Each mouse was genotyped at 10,346 single nucleotide polymorphisms (SNPs) that were coded as 0, 1 and 2. Here we use two continuous traits, body length (BL) and body mass index (BMI).

Pig data: The largest tabular dataset in our study is the pig data (Cleveland et al., 2012) which contains 3534 individuals with high-density genotypes and continuous phenotypes of five anonymized traits. After cleaning some missing data, we finally obtained 2314 samples, where each sample contains 52,843 SNPs. The data was anonymised by randomising the map order and recording the SNP identities.

Wheat data: The wheat dataset originates from CIMMYT's Global Wheat Program and is a publicly available in the BGLR package (Pérez & de Los Campos, 2014). It comprises 599 wheat lines from the CIMMYT Global Wheat Program evaluated in four international environments representing four basic agroclimatic regions (mega-environments). The wheat lines were genotyped using 1,447 Diversity Array Technology (DArT) markers. As a quality control, all the markers with a minor allele frequency below 0.05 were eliminated, and any missing genotypes were imputed once using samples from the marginal distribution of marker genotypes. Following these procedures, the dataset was reduced to 1,279 DArT markers.

Loblolly pine data: This data is derived from 32 parents representing a wide range of accessions from the Atlantic coastal plain, Florida, and the lower Gulf of the United States. Parents were crossed in a circular mating design with additional off-diagnal crosses, resulting in 70 full-sib families with an average of 13.5 individuals per family (Resende Jr et al., 2012; Baltunis et al., 2006). It was originally composed of 951 individuals from 61 families that was genotyped using an Illumina Infinium assay (Eckert et al., 2010). A subset of 4,853 SNPs (encoded as 0, 1, 2) were polymorphic and used in our study. There are totally 17 traits in the original data. We selected seven traits: rootnum (Trait 1), rootnumbin (Trait 2), c5c6 (Trait 3), density (Trait 4), lateWood%4 (Trait 5), lignin (Trait 6), and stiffnessTree (Trait 7) due to their high data quality. After cleaning the missing values, we finally got 806 samples and each sample contains 4,853 SNPs. The dataset was divided into three subsets—training, validation and testing - using the same percentage as other data used.

## 5    Implementation Details

**Training ResNets.** We trained ResNets on four real genomic datasets using various regularizers defined by the objective function with convex and non-convex $L_q$ regularization: $\mathcal{F}(\theta) = \mathcal{L}(\theta) + \lambda \sum_{j=1}^{p} |\theta_j|^q$, where $0 \leq q \leq \infty$. The parameters were trained with the closed-form proximal mappings presented in Appendix A. We explored a range of ResNet architectures aimed for 1D genomic data, including ResNet-5, ResNet-10, ResNet-15, ResNet-18, ResNet-20, ResNet-25. Each model consists of basic block modules with convolutional layers, batch normalization, and a final fully connected linear output layer. To incorporate regularization, we implemented a custom optimizer, which extends the standard PyTorch class. This optimizer combines an adaptive gradient step, similar to the Adam optimizer, with a subsequent proximal mapping step. Our implementation uses a momentum-like update with $\beta_1 = 0.1$ and $\beta_2 = 0.999$. After the gradient update, the optimizer applies a proximal operator to each weight, defined by a specific regularization function. During training, the validation mean squared error (MSE) was monitored closely. If the model failed to adequately capture the data complexity, increasing the size of the network was considered. In contrast, if overfitting persisted despite regularization efforts, a smaller network with fewer layers was evaluated. Additionally, hyperparameter tuning was conducted to determine optimal values for the learning rate, batch size and regularization parameters. All features were normalized to zero mean and unit variance prior to training.

**Training traditional sparse proximal models and LightGBM.** For the purpose of comparison, we evaluated proximal gradient descent linear regression models (PGDLM) with the same regularizers as for the ResNet models. We used Bayesian optimization (BO) to find the best regularization factor of the penalties (Frazier, 2018). In an earlier comparative evaluation of methods for GWP  (Fan & Waldmann, 2024), we found that LightGBM  (Ke et al., 2017) outperformed both traditional linear models and several different deep learning architectures. Hence, we decided to use it as a non-linear baseline method here as well. For the LightGBM, we used $L_1$ regularization to prevent the model from overfitting. The range for the regularization factor was set from 0.000001 to 0.1. Performance metrics and evaluation frameworks were aligned with our earlier work  (Fan et al., 2024), enabling a thorough comparison of results. For performance assessment, we computed the average test MSE, and distance correlation coefficient (dCor) (Székely et al., 2007) for all traits of each dataset. For the sparsity inducing regularizers, the optimal solution of the weights $\mathbf{w}^*$ for the problem (1) will be sparse. The sparsity was calculated as

$$\text{sparsity} = \frac{\text{The number of non-zeros entries of } \mathbf{w}^*}{\text{Dimension of } \mathbf{w}^*} = \frac{||\mathbf{w}^*||_0}{d}. \tag{7}$$

**Tuning parameters.** To achieve the best configurations of the ResNets, we utilized BO also here. The proposed approach involved initially dividing the data randomly into 5-fold cross-validation sets. The BO process was executed on the training/validation set, with model performance measured by the average validation MSE across the 5 folds. The BO search space for each ResNet configuration was defined for three key hyperparameters: (1) learning rate: continuous range from $10^{-4}$ to $10^{-2}$. (2) regularization parameter: continuous range from $10^{-3}$ to $10^2$. (3) batch size: selected from a discrete set of choices: 32, 64, or 128. To improve the computation efficiency, models were executed in parallel across each fold, with the test MSE calculated independently. The BO process was terminated by monitoring the improvement in model performance over the iterations. To ensure diverse exploration of the parameter space, the Tree-structured Parzen Estimator (TPE) method was used within BO to leverage the model's inherent stochasticity while integrating new recommendations  (Frazier, 2018). The BO was implemented using the Hyperopt library (Bergstra et al., 2013). Figure 3 shows the training losses across the four datasets for the different regularizers.

**Parallel computing.** All models were implemented using the PyTorch framework, and experiments were run on five NVIDIA GPUs to accelerate training. Parallelization across multiple folds was employed during cross-validation to enhance computational efficiency. The final model configurations were determined by running BO for a maximum of 100 iterations, with convergence considered achieved when the improvement in validation MSE was less than $10^{-5}$ for five consecutive iterations. The code for all the algorithms are available online at: https://anonymous.4open.science/r/Adaptive-gradient-methods.

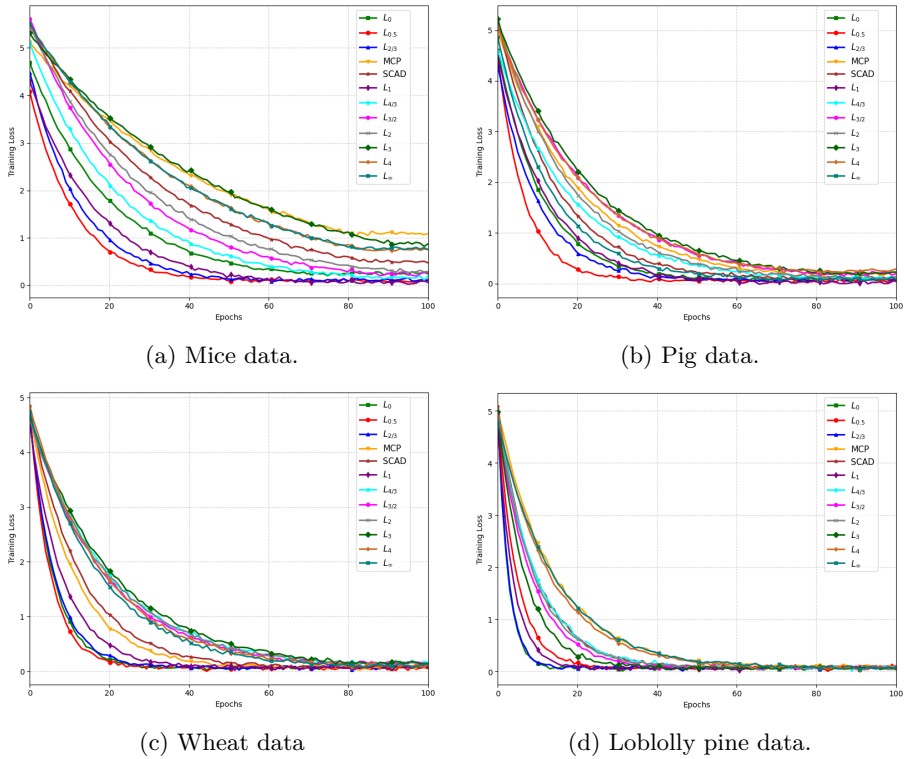

(a) Mice data.

(b) Pig data.

(c) Wheat data

(d) Loblolly pine data.

Figure 3: Training loss curves demonstrating various regularization effects on four datasets: ResNet-18 for the mice and pig data and ResNet-15 for the wheat and loblolly pine data.

## 6 Results and Discussion

### 6.1 Comparison of Adaptive Proximal Methods on Different Datasets

**Mice data:** From Table 1, it is evident that the choice of regularizer significantly impacts performance. The $L_{\frac{1}{2}}$ regularizer consistently outperformed all other methods, achieving the lowest test MSE and highest distance correlation (dCor) values. For Trait 1, it yielded an MSE of $0.133 \pm 0.002$ and a dCor of $0.733 \pm 0.001$. Similarly, for Trait 2, it produced the lowest MSE of $0.134 \pm 0.005$ and the highest dCor of $0.355 \pm 0.004$. This demonstrates that the $L_{\frac{1}{2}}$ penalty is highly effective for this genomic dataset. The performance of other regularizers varied, but the non-convex regularizers like MCP and SCAD, and those with higher $L_q$ norms (e.g., $L_{\frac{3}{2}}$, $L_3$, $L_\infty$), performed worse than $L_{\frac{1}{2}}$ and $L_2$. For instance, MCP and SCAD yielded higher MSE and lower dCor values for both traits compared to $L_{\frac{1}{2}}$. The ResNet-18 models with any of the tested regularizers significantly outperformed the baseline methods, PGDLM and LightGBM. The PGDLM with $L_{\frac{1}{2}}$ baseline had a test MSE of $0.177 \pm 0.005$ for Trait 1, which is notably higher than the $0.133 \pm 0.002$ achieved by the ResNet-18 with the same regularizer. Similarly, LightGBM with $L_1$ performed the worst among all tested models, with a test MSE of $0.194 \pm 0.006$ for Trait 1 and $0.203 \pm 0.007$ for Trait 2. This highlights the substantial advantage of the deep learning-based ResNet approach over both traditional linear models and tree-based methods for this specific genomic prediction task.

**Pig data:** The pig dataset is the largest among the datasets used in this study, comprising five traits. As shown in Table 2, we observed a significant performance improvement in test MSE and distance correlation (dCor) when increasing the ResNet depth from 5 to 18 layers across all five traits. For instance, using the $L_{\frac{1}{2}}$ regularizer, the test MSE for Trait 1 decreased from $0.140 \pm 0.003$ (ResNet-5) to $0.129 \pm 0.005$ (ResNet-18). This trend highlights that deeper networks are better at learning complex feature representations in larger genomic datasets, leading to superior prediction performance. However, increasing the network depth beyond 18 layers offered only marginal gains. For example, for Trait 4 with the $L_{\frac{1}{2}}$ regularizer, the MSE

Table 1: Performance comparison among ResNet-18 with different regularizers and other baselines on the mice data. Test MSE (mean with stddev) and dCor (mean with stddev). **Bold** values indicate the regularizer yielding the lowest test MSE and the highest distance correlation values. Only the best performing regularizer for PGDLM is presented.

| Method | Regularizer | Trait 1 | | Trait 2 | |
|---|---|---|---|---|---|
| | | **MSE** | **dCor** | **MSE** | **dCor** |
| | $L_0$ | 0.138 (0.005) | 0.573 (0.003) | 0.141 (0.003) | 0.272 (0.002) |
| | $L_{\frac{1}{2}}$ | **0.133 (0.002)** | **0.733 (0.001)** | **0.134 (0.005)** | **0.355 (0.004)** |
| | $L_{\frac{2}{3}}$ | 0.134 (0.007) | 0.701 (0.011) | 0.136 (0.005) | 0.301 (0.001) |
| | MCP | 0.142 (0.013) | 0.551 (0.001) | 0.146 (0.003) | 0.264 (0.002) |
| | SCAD | 0.144 (0.003) | 0.542 (0.007) | 0.147 (0.006) | 0.261 (0.005) |
| ResNet-18 | $L_1$ | 0.136 (0.005) | 0.637 (0.003) | 0.139 (0.008) | 0.280 (0.004) |
| | $L_{\frac{4}{3}}$ | 0.136 (0.014) | 0.592 (0.002) | 0.139 (0.013) | 0.277 (0.002) |
| | $L_{\frac{3}{2}}$ | 0.136 (0.007) | 0.554 (0.003) | 0.140 (0.006) | 0.265 (0.001) |
| | $L_2$ | 0.137 (0.017) | 0.642 (0.001) | 0.141 (0.007) | 0.282 (0.003) |
| | $L_3$ | 0.139 (0.005) | 0.650 (0.005) | 0.143 (0.013) | 0.296 (0.001) |
| | $L_4$ | 0.139 (0.010) | 0.673 (0.004) | 0.142 (0.007) | 0.297 (0.003) |
| | $L_\infty$ | 0.138 (0.003) | 0.605 (0.001) | 0.141 (0.005) | 0.277 (0.005) |
| PGDLM | $L_{\frac{1}{2}}$ | 0.177 (0.005) | 0.564 (0.005) | 0.185 (0.014) | 0.269 (0.002) |
| LightGBM | $L_1$ | 0.194 (0.006) | 0.550 (0.003) | 0.203 (0.007) | 0.262 (0.002) |

showed a small improvement from $0.131 \pm 0.004$ (ResNet-18) to $0.130 \pm 0.001$ (ResNet-25), demonstrating a point of diminishing returns.

From Table 2, it is clear that the $L_{\frac{1}{2}}$ regularizer consistently delivered the best performance on ResNet-18 for all five traits, achieving the lowest test MSE and highest dCor. For instance, for Trait 4, it produced the lowest MSE of $0.126 \pm 0.001$ and the highest dCor of $0.982 \pm 0.004$. This shows that the pig data, with its larger size, benefits more from the specific sparsity and smoothness properties induced by the $L_{\frac{1}{2}}$ penalty. While performance differences among regularizers narrowed in deeper networks, the $L_{\frac{1}{2}}$ penalty remained the most effective. We can clearly see that all ResNet-18 models with regularization outperformed the baseline methods. The PGDLM with $L_{\frac{1}{2}}$ model and LightGBM showed significantly higher MSE and lower dCor values. This underscores the substantial advantage of our deep learning-based ResNet approach over both traditional linear models and tree-based methods for this specific genomic prediction task, particularly in handling large and complex datasets where non-linear feature interactions are more pronounced.

**Wheat data:** The wheat dataset, which consists of four distinct traits, was analyzed using various regularizers applied to ResNet architectures of different depths. As detailed in Table 3, among the ResNet architectures we tested (ResNet-5, ResNet-10, ResNet-15, ResNet-18, ResNet-20, and ResNet-25), ResNet-15 yielded the best overall performance, demonstrating an optimal balance between model complexity and predictive power for this dataset. Deeper models like ResNet-18 and ResNet-25 did not offer significant performance improvements.

The results show that the $L_{\frac{1}{2}}$ regularizer consistently outperformed all other regularization methods, achieving the lowest test MSE and highest distance correlation (dCor) values across all four traits. For Trait 1, it yielded a test MSE of $0.132 \pm 0.002$; for Trait 2, $0.131 \pm 0.002$; for Trait 3, $0.130 \pm 0.003$; and for Trait 4, $0.130 \pm 0.002$. This consistent superiority suggests that the $L_{\frac{1}{2}}$ penalty provides effective model generalization and is particularly well-suited for the complex, high-dimensional data in this genomic prediction task. While the performance differences between the regularizers were not large, several obvious observations can be made. Regularizers such as SCAD and MCP, and those with higher $L_q$ norms like $L_{\frac{4}{3}}$ and $L_{\frac{3}{2}}$, consistently resulted in higher test MSEs compared to $L_{\frac{1}{2}}$ and $L_2$. For instance, SCAD and MCP yielded test MSEs of $0.144 \pm 0.005$ and $0.143 \pm 0.005$, respectively, for Trait 1, which are notably higher than the $0.132 \pm 0.002$ achieved by $L_{\frac{1}{2}}$. This shows the importance of selecting the regularizer to the specific characteristics of the data. Furthermore, the performance of $L_0$, $L_{\frac{2}{3}}$, and $L_{\frac{4}{3}}$ was similar across traits and slightly inferior to $L_{\frac{1}{2}}$, highlighting the robustness and consistent effectiveness of the $L_{\frac{1}{2}}$ penalty.

Table 2: Performance comparison among ResNet-18 with different regularizers and other baselines on the pig data. Test MSE (mean with stddev) and dCor (mean with stddev). **Bold** values indicate the regularizer yielding the lowest test MSE and the highest distance correlation values. Only the best performing regularizer for PGDLM is presented.

| Method | Regularizer | Trait 1 | | Trait 2 | | Trait 3 | | Trait 4 | | Trait 5 | |
|---|---|---|---|---|---|---|---|---|---|---|---|
| | | MSE | dCor | MSE | dCor | MSE | dCor | MSE | dCor | MSE | dCor |
| | $L_0$ | 0.136 (0.003) | 0.801 (0.002) | 0.137 (0.001) | 0.753 (0.004) | 0.136 (0.002) | 0.793 (0.003) | 0.132 (0.004) | 0.772 (0.002) | 0.144 (0.003) | 0.825 (0.005) |
| | $L_{\frac{1}{2}}$ | **0.129 (0.005)** | **0.901 (0.003)** | **0.130 (0.004)** | **0.924 (0.001)** | **0.129 (0.004)** | **0.910 (0.005)** | **0.126 (0.001)** | **0.982 (0.004)** | **0.133 (0.002)** | **0.941 (0.001)** |
| | $L_{\frac{2}{3}}$ | 0.131 (0.002) | 0.893 (0.002) | 0.132 (0.005) | 0.904 (0.004) | 0.131 (0.003) | 0.877 (0.002) | 0.127 (0.003) | 0.950 (0.001) | 0.135 (0.001) | 0.902 (0.004) |
| ResNet-18 | MCP | 0.137 (0.004) | 0.771 (0.003) | 0.138 (0.003) | 0.721 (0.003) | 0.137 (0.004) | 0.765 (0.002) | 0.134 (0.001) | 0.752 (0.004) | 0.144 (0.003) | 0.791 (0.004) |
| | SCAD | 0.138 (0.002) | 0.766 (0.005) | 0.139 (0.004) | 0.713 (0.003) | 0.138 (0.002) | 0.767 (0.006) | 0.134 (0.005) | 0.749 (0.003) | 0.145 (0.003) | 0.782 (0.012) |
| | $L_1$ | 0.135 (0.001) | 0.843 (0.002) | 0.136 (0.002) | 0.821 (0.007) | 0.135 (0.003) | 0.827 (0.008) | 0.132 (0.002) | 0.816 (0.002) | 0.143 (0.001) | 0.852 (0.009) |
| | $L_{\frac{4}{3}}$ | 0.135 (0.002) | 0.802 (0.003) | 0.136 (0.004) | 0.787 (0.002) | 0.135 (0.005) | 0.806 (0.002) | 0.132 (0.002) | 0.784 (0.003) | 0.143 (0.001) | 0.830 (0.001) |
| | $L_{\frac{3}{2}}$ | 0.136 (0.002) | 0.793 (0.002) | 0.137 (0.002) | 0.743 (0.004) | 0.136 (0.002) | 0.772 (0.004) | 0.132 (0.003) | 0.763 (0.004) | 0.144 (0.002) | 0.813 (0.002) |
| | $L_2$ | 0.133 (0.003) | 0.863 (0.003) | 0.134 (0.003) | 0.854 (0.003) | 0.133 (0.004) | 0.847 (0.003) | 0.130 (0.001) | 0.852 (0.003) | 0.140 (0.005) | 0.861 (0.003) |
| | $L_3$ | 0.132 (0.003) | 0.841 (0.002) | 0.133 (0.002) | 0.873 (0.004) | 0.132 (0.003) | 0.853 (0.003) | 0.128 (0.001) | 0.872 (0.006) | 0.136 (0.003) | 0.870 (0.002) |
| | $L_4$ | 0.132 (0.002) | 0.886 (0.003) | 0.132 (0.003) | 0.873 (0.005) | 0.132 (0.004) | 0.871 (0.004) | 0.127 (0.003) | 0.920 (0.002) | 0.136 (0.001) | 0.881 (0.002) |
| | $L_\infty$ | 0.134 (0.002) | 0.854 (0.002) | 0.135 (0.001) | 0.779 (0.004) | 0.134 (0.002) | 0.813 (0.002) | 0.131 (0.003) | 0.784 (0.002) | 0.142 (0.001) | 0.791 (0.004) |
| PGDLM | $L_{\frac{1}{2}}$ | 0.177 (0.005) | 0.802 (0.002) | 0.185 (0.014) | 0.801 (0.004) | 0.179 (0.010) | 0.811 (0.004) | 0.171 (0.002) | 0.802 (0.005) | 0.188 (0.008) | 0.837 (0.003) |
| LightGBM | $L_1$ | 0.194 (0.006) | 0.692 (0.003) | 0.203 (0.007) | 0.653 (0.007) | 0.198 (0.011) | 0.711 (0.003) | 0.189 (0.005) | 0.633 (0.007) | 0.211 (0.009) | 0.681 (0.005) |

Table 3: Performance comparison among ResNet-15 with different regularizers and other baselines on the wheat data. Test MSE (mean with stddev) and dCor (mean with stddev). **Bold** values indicate the regularizer yielding the lowest test MSE and the highest distance correlation values. Only the best performing regularizer for PGDLM is presented.

| Method | Regularizer | Trait 1 | | Trait 2 | | Trait 3 | | Trait 4 | |
|---|---|---|---|---|---|---|---|---|---|
| | | MSE | dCor | MSE | dCor | MSE | dCor | MSE | dCor |
| ResNet-15 | $L_0$ | 0.142 (0.001) | 0.719 (0.004) | 0.137 (0.003) | 0.761 (0.001) | 0.134 (0.005) | 0.619 (0.002) | 0.135 (0.002) | 0.742 (0.002) |
| | $L_{\frac{1}{5}}$ | **0.132 (0.002)** | **0.854 (0.003)** | **0.131 (0.002)** | **0.853 (0.001)** | **0.130 (0.003)** | **0.737 (0.002)** | **0.130 (0.002)** | **0.819 (0.004)** |
| | $L_{\frac{2}{3}}$ | 0.135 (0.001) | 0.816 (0.003) | 0.132 (0.005) | 0.850 (0.001) | 0.132 (0.003) | 0.722 (0.002) | 0.132 (0.002) | 0.792 (0.002) |
| | MCP | 0.143 (0.005) | 0.765 (0.004) | 0.137 (0.003) | 0.742 (0.004) | 0.135 (0.002) | 0.592 (0.005) | 0.136 (0.002) | 0.721 (0.004) |
| | SCAD | 0.144 (0.005) | 0.761 (0.002) | 0.138 (0.004) | 0.728 (0.005) | 0.136 (0.001) | 0.591 (0.002) | 0.137 (0.002) | 0.720 (0.002) |
| | $L_1$ | 0.140 (0.003) | 0.743 (0.006) | 0.136 (0.001) | 0.807 (0.004) | 0.134 (0.001) | 0.621 (0.002) | 0.134 (0.003) | 0.771 (0.003) |
| | $L_{\frac{4}{3}}$ | 0.142 (0.002) | 0.724 (0.002) | 0.136 (0.003) | 0.772 (0.002) | 0.134 (0.004) | 0.621 (0.003) | 0.134 (0.001) | 0.773 (0.002) |
| | $L_{\frac{3}{2}}$ | 0.142 (0.001) | 0.716 (0.004) | 0.137 (0.001) | 0.756 (0.003) | 0.134 (0.002) | 0.608 (0.002) | 0.135 (0.003) | 0.753 (0.005) |
| | $L_2$ | 0.139 (0.004) | 0.757 (0.002) | 0.135 (0.003) | 0.822 (0.002) | 0.133 (0.002) | 0.653 (0.001) | 0.134 (0.001) | 0.776 (0.003) |
| | $L_3$ | 0.136 (0.002) | 0.763 (0.003) | 0.133 (0.003) | 0.836 (0.005) | 0.133 (0.002) | 0.704 (0.002) | 0.134 (0.001) | 0.774 (0.003) |
| | $L_4$ | 0.135 (0.005) | 0.775 (0.002) | 0.132 (0.001) | 0.841 (0.002) | 0.133 (0.002) | 0.703 (0.002) | 0.133 (0.002) | 0.784 (0.001) |
| | $L_\infty$ | 0.140 (0.002) | 0.732 (0.002) | 0.135 (0.001) | 0.791 (0.002) | 0.133 (0.003) | 0.638 (0.001) | 0.134 (0.003) | 0.765 (0.003) |
| PGDLM | $L_{\frac{1}{2}}$ | 0.177 (0.005) | 0.501 (0.007) | 0.185 (0.014) | 0.483 (0.006) | 0.179 (0.010) | 0.571 (0.006) | 0.171 (0.012) | 0.560 (0.001) |
| LightGBM | $L_1$ | 0.194 (0.006) | 0.475 (0.003) | 0.203 (0.007) | 0.461 (0.002) | 0.198 (0.011) | 0.503 (0.004) | 0.189 (0.005) | 0.511 (0.007) |

**Loblolly pine data:** The analysis of the loblolly pine dataset, which includes seven traits, revealed that **ResNet-15** provided the best overall performance among the architectures we tested (ResNet-10, ResNet-15, ResNet-18, ResNet-20, and ResNet-25). This finding is consistent with our results for the wheat data, and suggests that an intermediate network depth strikes the right balance between model capacity and generalization for genomic prediction tasks. As shown in Table 4, the $L_{\frac{1}{2}}$ regularizer consistently achieved the lowest test MSE and highest dCor across all traits. The test MSE for $L_{\frac{1}{2}}$ ranged from $0.139 \pm 0.005$ (Trait 4) to $0.145 \pm 0.006$ (Trait 7), outperforming all other regularizers. This outcome reinforces the prior observations from the mice and wheat datasets, highlighting the superior effectiveness of the $L_{\frac{1}{2}}$ penalty in this context.

In comparison, the convex regularizers $L_1$ and $L_2$ performed moderately well, with MSE values close to, but generally higher than, those of $L_{\frac{1}{2}}$. For example, for Trait 1, the $L_2$ regularizer yielded an MSE of $0.145 \pm 0.011$ compared to $0.141 \pm 0.007$ for $L_{\frac{1}{2}}$. This small but consistent performance gap suggests that fractional penalties offer a slight advantage by encouraging sparsity without being overly restrictive. Other regularizers, including the non-convex MCP and SCAD, and the higher-order $L_p$ norms ($L_3$, $L_4$, and $L_\infty$), performed notably worse. SCAD yielded some of the highest MSE values, such as $0.160 \pm 0.002$ for Trait 7, with MCP showing a similar degradation in predictive accuracy. The consistent underperformance of these regularizers across the mice, wheat, and pine datasets indicates that they may not be well-suited for deeper ResNet architectures in genomic prediction. The higher-order $L_p$ norms also underperformed, yielding test MSEs in the range of $0.146 - 0.155$. For instance, $L_3$ resulted in an MSE of $0.155 \pm 0.003$ for Trait 7, and $L_\infty$ gave $0.153 \pm 0.007$ for the same trait. This highlights the importance of using penalties that effectively encourage sparsity without losing critical information, a balance that the $L_{\frac{1}{2}}$ regularizer appears to strike exceptionally well.

## 6.2 Adaptive Proximal Methods vs. Traditional Methods

### 6.2.1 Model Depth and Complexity

**Model depth:** For the adaptive proximal methods with ResNets, increasing network depth helps capture more complex relationships between features, as seen with ResNet-18 often yielding the best results on the mice and pig data and with ResNet-15 obtaining the smallest MSE on the wheat and pine data. This depth enables the model to learn more intricate patterns in the data that traditional methods might miss. Regularizers such as $L_{\frac{1}{2}}$ can adapt to the depth of the network, resulting in improved performance with increasing depth. The improvement in MSE increases with ResNet depth across most regularizers, suggesting that deeper networks enhance performance. In contrast, the traditional PGDLM lack the capacity to learn hierarchical or complex interactions between features. LightGBM offers competitive performance on several genomic datasets due to its ability to model nonlinear relationships and handle high-dimensional inputs. However, unlike ResNets, LightGBM does not exploit hierarchical feature composition across multiple layers. Its performance is often strong for individual traits, but it struggles to generalize consistently across multiple correlated traits.

**Complexity:** The computational complexity for traditional sparse proximal methods is $O(n \cdot p + p)$ per iteration, where $n$ represents the number of samples and $p$ the number of features. In comparison, adaptive proximal gradient methods utilizing ResNets have a higher computational complexity due to the deeper network architecture and additional complexity introduced by the proximal operator for regularization. The complexity of adaptive gradient methods using ResNets is influenced by several factors, such as network depth, width, and the specific nature of the applied regularization. Each forward pass in a ResNet is computationally intensive, particularly for deeper networks, and the proximal gradient step further adds complexity based on the number of features and the nature of the proximal operator. The overall computational complexity of adaptive proximal gradient methods is $O(L \cdot n \cdot d^2 + n \cdot p + p)$, where $L$ is the number of layers, $K$ is the number of residual blocks per layer, and $d$ is the number of blocks per layer. It is obvious that adaptive proximal methods have much higher computational complexity than traditional sparse proximal methods. LightGBM, in contrast, has per-iteration complexity dominated by tree construction and leaf-wise splitting, typically $O(n \cdot \log n)$ per tree, making it more computationally efficient than deep ResNets. However, this efficiency comes at the cost of reduced expressiveness, since tree-based models do not scale in representational

Table 4: Performance comparison among ResNet-15 with different regularizers and other baselines on the loblolly pine data. Test MSE (mean with stddev) and dCor (mean with stddev). **Bold** values indicate the regularizer yielding the lowest test MSE and the highest distance correlation values. Only the best performing regularizer for PGDLM is presented.

| Method | Regularizer | Trait 1 MSE | Trait 1 dCor | Trait 2 MSE | Trait 2 dCor | Trait 3 MSE | Trait 3 dCor | Trait 4 MSE | Trait 4 dCor | Trait 5 MSE | Trait 5 dCor | Trait 6 MSE | Trait 6 dCor | Trait 7 MSE | Trait 7 dCor |
|---|---|---|---|---|---|---|---|---|---|---|---|---|---|---|---|
| ResNet-15 | $L_0$ | 0.146 (0.003) | 0.735 (0.003) | 0.146 (0.007) | 0.461 (0.002) | 0.145 (0.006) | 0.508 (0.003) | 0.143 (0.007) | 0.531 (0.003) | 0.145 (0.009) | 0.551 (0.003) | 0.148 (0.005) | 0.681 (0.003) | 0.148 (0.004) | 0.702 (0.003) |
| | $L_{\frac{1}{2}}$ | **0.141** (0.007) | **0.883** (0.003) | **0.145** (0.007) | **0.521** (0.003) | **0.140** (0.005) | **0.656** (0.001) | **0.139** (0.005) | **0.637** (0.002) | **0.140** (0.005) | **0.617** (0.006) | **0.143** (0.009) | **0.753** (0.004) | **0.145** (0.006) | **0.791** (0.003) |
| | $L_{\frac{2}{3}}$ | 0.143 (0.004) | 0.833 (0.002) | 0.145 (0.007) | 0.504 (0.002) | 0.143 (0.007) | 0.617 (0.003) | 0.142 (0.007) | 0.622 (0.003) | 0.143 (0.006) | 0.595 (0.008) | 0.145 (0.003) | 0.743 (0.001) | 0.146 (0.002) | 0.758 (0.003) |
| | MCP | 0.151 (0.011) | 0.691 (0.003) | 0.156 (0.017) | 0.442 (0.009) | 0.150 (0.007) | 0.473 (0.003) | 0.146 (0.003) | 0.515 (0.001) | 0.148 (0.007) | 0.532 (0.002) | 0.155 (0.004) | 0.665 (0.003) | 0.157 (0.002) | 0.681 (0.001) |
| | SCAD | 0.154 (0.007) | 0.690 (0.002) | 0.158 (0.002) | 0.442 (0.004) | 0.151 (0.002) | 0.470 (0.003) | 0.148 (0.003) | 0.509 (0.002) | 0.149 (0.006) | 0.531 (0.004) | 0.156 (0.007) | 0.659 (0.002) | 0.160 (0.002) | 0.681 (0.002) |
| | $L_1$ | 0.146 (0.002) | 0.761 (0.002) | 0.147 (0.004) | 0.471 (0.002) | 0.145 (0.006) | 0.543 (0.004) | 0.144 (0.007) | 0.542 (0.002) | 0.144 (0.003) | 0.573 (0.003) | 0.146 (0.002) | 0.703 (0.002) | 0.148 (0.017) | 0.736 (0.002) |
| | $L_{\frac{4}{3}}$ | 0.147 (0.016) | 0.740 (0.001) | 0.149 (0.003) | 0.465 (0.001) | 0.146 (0.005) | 0.511 (0.002) | 0.145 (0.002) | 0.531 (0.002) | 0.146 (0.016) | 0.561 (0.001) | 0.147 (0.007) | 0.686 (0.003) | 0.150 (0.002) | 0.717 (0.001) |
| | $L_{\frac{3}{2}}$ | 0.148 (0.002) | 0.712 (0.003) | 0.151 (0.001) | 0.450 (0.003) | 0.148 (0.007) | 0.477 (0.005) | 0.146 (0.007) | 0.523 (0.003) | 0.147 (0.002) | 0.540 (0.003) | 0.149 (0.016) | 0.677 (0.001) | 0.153 (0.003) | 0.692 (0.005) |
| | $L_2$ | 0.145 (0.011) | 0.773 (0.002) | 0.147 (0.009) | 0.482 (0.010) | 0.144 (0.007) | 0.566 (0.002) | 0.143 (0.006) | 0.545 (0.002) | 0.144 (0.015) | 0.581 (0.001) | 0.146 (0.002) | 0.712 (0.004) | 0.147 (0.004) | 0.736 (0.004) |
| | $L_3$ | 0.151 (0.003) | 0.790 (0.007) | 0.154 (0.003) | 0.491 (0.005) | 0.150 (0.015) | 0.581 (0.002) | 0.146 (0.007) | 0.559 (0.002) | 0.148 (0.003) | 0.591 (0.003) | 0.153 (0.017) | 0.727 (0.007) | 0.155 (0.003) | 0.742 (0.002) |
| | $L_4$ | 0.144 (0.001) | 0.812 (0.012) | 0.146 (0.007) | 0.502 (0.002) | 0.143 (0.008) | 0.607 (0.001) | 0.145 (0.002) | 0.613 (0.005) | 0.144 (0.007) | 0.595 (0.004) | 0.144 (0.011) | 0.732 (0.006) | 0.147 (0.003) | 0.749 (0.003) |
| | $L_\infty$ | 0.149 (0.005) | 0.753 (0.002) | 0.151 (0.003) | 0.470 (0.002) | 0.149 (0.002) | 0.543 (0.003) | 0.146 (0.004) | 0.540 (0.003) | 0.147 (0.011) | 0.565 (0.003) | 0.149 (0.002) | 0.692 (0.005) | 0.153 (0.007) | 0.725 (0.002) |
| PGDLM | $L_{\frac{1}{2}}$ | 0.177 (0.005) | 0.577 (0.005) | 0.185 (0.014) | 0.502 (0.002) | 0.179 (0.010) | 0.473 (0.004) | 0.171 (0.012) | 0.520 (0.004) | 0.188 (0.008) | 0.548 (0.006) | 0.175 (0.007) | 0.463 (0.006) | 0.182 (0.009) | 0.691 (0.004) |
| LightGBM | $L_1$ | 0.194 (0.006) | 0.561 (0.002) | 0.203 (0.007) | 0.491 (0.002) | 0.198 (0.011) | 0.464 (0.002) | 0.189 (0.005) | 0.493 (0.002) | 0.211 (0.009) | 0.533 (0.005) | 0.199 (0.012) | 0.461 (0.001) | 0.205 (0.010) | 0.661 (0.003) |

Table 5: Sparsity (%) comparison for ResNet-18 (mice and pig) and ResNet-15 (wheat and loblolly pine) with non-convex regularizers across four genomic datasets.

| Regularizer | Mice data | Pig data | Wheat data | Loblolly pine data |
|---|---|---|---|---|
| $L_0$ | 15.09 | 15.76 | 14.73 | 14.88 |
| $L_{\frac{1}{2}}$ | 13.17 | 13.26 | 12.09 | 12.15 |
| $L_{\frac{2}{3}}$ | 14.13 | 14.55 | 13.81 | 13.87 |
| MCP | 14.71 | 15.01 | 13.33 | 13.81 |
| SCAD | 15.03 | 15.76 | 14.11 | 14.53 |
| $L_1$ | 12.19 | 13.62 | 12.09 | 13.15 |

power with added "depth" in the same sense as ResNets. Thus, while LightGBM is computationally attractive, adaptive proximal ResNets achieve superior predictive performance at the expense of higher complexity and computational demands.

### 6.3 Comparison of Sparsity and Computing Efficiency

Table 5 highlights the joint effects of different regularizers on sparsity. We can clearly see that across all four datasets, the $L_{\frac{1}{2}}$ penalty consistently achieved the most favorable trade-off. It produced the lowest or near-lowest test MSE, the strongest distance correlations, and compact models with sparsity levels around 12 - 13%. In contrast, convex baselines such as $L_1$ and $L_2$ induced moderate sparsity but were systematically outperformed in both accuracy and correlation. Higher-order convex penalties ($L_3$, $L_4$) occasionally improved correlation relative to $L_1$ and $L_2$, but they did not match the predictive efficiency of fractional regularization. Non-convex penalties such as MCP and SCAD often drove sparsity above 14 - 15%, but this came at the cost of unstable generalization, with consistently higher errors and weaker correlations across all datasets.

We also noticed that there are significant computing demand differences among ResNets. The training time is reflected in dataset size (20 - 30 minutes for wheat, around 200 minutes for pigs), but the experiments showed little variation across regularizers with the same model depth. This indicates that the performance gains of fractional regularization were not accompanied by additional computational burden. These results demonstrate that fractional penalties, and in particular $L_{\frac{1}{2}}$, provide a stable and scalable mechanism for genomic prediction since they yield sparse and accurate models.

## 7 Conclusion

This study demonstrates that adaptive proximal methods with ResNets provide superior genomic prediction performance across diverse datasets. Deeper architectures, such as ResNet-18 and ResNet-15, consistently improved predictive accuracy, while the $L_{\frac{1}{2}}$ regularizer yielded the lowest test MSEs and highest distance correlations. Compared to traditional sparse proximal methods and LightGBM, ResNet-based adaptive proximal methods achieved more accurate and consistent predictions, highlighting their robustness and generalizability for high-dimensional, multi-trait genomic datasets.

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

## Appendix

## A    Proximal Mappings of The Regularization Functions

Here we present the proximal operators for the regularizers $L_0$ to $L_\infty$ and how they are combined with the Adam preconditioners presented in Section 3. We also consider the cases of MCP and SCAD regularization. Recall that the minimization problem for $L_q$ regularizers is

$$\arg\min_\theta \{\frac{1}{2}\left\|\theta - \hat{\theta}_t\right\|^2_{C_t+\delta I} + \lambda \sum_{j=1}^{p} |\theta_j|^q\}, \tag{8}$$

for which a solution is obtained by iterating

$$\hat{\theta}_t = \theta_t - \alpha_t (C_t + \delta I)^{-1} m_t,$$
$$\theta_{t+1} = \text{prox}_{\alpha_t \lambda \mathcal{R}(\cdot)}^{C_t + \delta I}(\hat{\theta}_t). \tag{9}$$

By denoting the $i$-th coordinate of $\theta_t$ as $\theta_{t,i}$ and the $i$-th diagonal entry of the preconditioner $[C_t]_{ii}$ as $C_{t,i}$, the formulation (9) is coordinate-wise decomposable and the preconditioner matrix $C_t$ is diagonal, so we get

$$\theta_{t+1,i} = \arg\min_{\theta_i} \{ \frac{1}{2}(C_{t,i} + \delta I)(\theta_i - \hat{\theta}_{t,i})^2 + \alpha_t \lambda |\theta_i|^q \}$$
$$= \arg\min_{\theta_i} \{ (\theta_i - \hat{\theta}_{t,i})^2 + \frac{2\alpha_t \lambda}{C_{t,i} + \delta I} |\theta_i|^q \}, \tag{10}$$

where

$$\hat{\theta}_{t,i} = \theta_{t,i} - \alpha_t \frac{m_{t,i}}{C_{t,i} + \delta I}. \tag{11}$$

## A.1  $L_0$ Regularization

When $q = 0$ in Equation (5), the regularization corresponds to the $L_0$ penalty function, which penalizes the number of non-zero entries of the parameter vector. The $L_0$ penalty is non-convex and promotes unbiased sparsity in the estimated parameters $\hat{\theta}$. In this case, the closed-form proximal operator can be obtained via hard-thresholding given by

$$\text{prox}_{\lambda \mathcal{R}(\cdot)}(\theta) = \begin{cases} \{0\}, & |\theta| < \sqrt{2\lambda}, \\ \{0, \theta\}, & |\theta| = \sqrt{2\lambda}, \\ \{\theta\}, & |\theta| > \sqrt{2\lambda}. \end{cases} \tag{12}$$

The closed-form proximal mappings given above for $L_0$ regularization can be combined with the preconditioner $C_{t,i}$ which leads to the following update rule  (Yun et al., 2021):

$$\theta_{t+1,i} = \begin{cases} \hat{\theta}_{t,i}, & \text{if } \left|\hat{\theta}_{t,i}\right| > \sqrt{\frac{2\alpha_t \lambda}{C_{t,i} + \delta I}}, \\ 0, & \text{if } \left|\hat{\theta}_{t,i}\right| < \sqrt{\frac{2\alpha_t \lambda}{C_{t,i} + \delta I}}, \\ \left\{0, \hat{\theta}_{t,i}\right\}, & \text{if } \left|\hat{\theta}_{t,i}\right| = \sqrt{\frac{2\alpha_t \lambda}{C_{t,i} + \delta I}}. \end{cases} \tag{13}$$

## A.2  $L_{\frac{1}{2}}$ Regularization

For $q = \frac{1}{2}$, the penalty corresponds to the $L_{\frac{1}{2}}$ quasinorm which encourages sparsity similarly to the $L_0$ penalty, but with smoother behavior near zero. This form of non-convex regularization applies shrinkage on the regression coefficients, but yields more sparse solutions than the $L_1$ regularizer. According to  (Cao et al., 2013), the proximal mapping is as follows:

$$\text{prox}_{\lambda \mathcal{R}(\cdot)}(\theta) = \begin{cases} \frac{2}{3}|\theta|\left(1 + \cos\left(\frac{2}{3}\pi - \frac{2}{3}\varphi_\lambda(\theta)\right)\right), & \text{if } \theta > p(\lambda), \\ 0, & \text{if } |\theta| \leq p(\lambda) \\ -\frac{2}{3}|\theta|\left(1 + \cos\left(\frac{2}{3}\pi - \frac{2}{3}\varphi_\lambda(\theta)\right)\right), & \text{if } \theta < p(\lambda) \end{cases} \tag{14}$$

where $\varphi_\lambda(\theta) = \arccos\left(\frac{\lambda}{4}\left(\frac{\theta}{3}\right)^{-\frac{3}{2}}\right)$ and $p(\lambda) = \frac{\sqrt[3]{54}}{4}(\lambda)^{\frac{2}{3}}$. Combined with the preconditioner, the $i$-th coordinate update becomes:

$$\theta_{t+1,i} = \begin{cases} \frac{2}{3}\left|\hat{\theta}_{t,i}\right|(1 + \cos(\frac{2}{3}\pi - \frac{2}{3}\varphi_\lambda(\hat{\theta}_{t,i}))), & \text{if } \hat{\theta}_{t,i} > p(\lambda) \\ 0, & \text{if } \left|\hat{\theta}_{t,i}\right| \leq p(\lambda) \\ -\frac{2}{3}\left|\hat{\theta}_{t,i}\right|(1 + \cos(\frac{2}{3}\pi - \frac{2}{3}\varphi_\lambda(\hat{\theta}_{t,i}))), & \text{if } \hat{\theta}_{t,i} < -p(\lambda) \end{cases} \tag{15}$$

where

$$\varphi_\lambda(\hat{\theta}_{t,i}) = \arccos\left(\frac{\alpha_t \lambda}{4(C_{t,i} + \delta I)}\left(\frac{|\hat{\theta}_{t,i}|}{3}\right)^{-\frac{3}{2}}\right), \quad p(\lambda) = \frac{\sqrt[3]{54}}{4}\left(\frac{2\alpha_t \lambda}{C_{t,i} + \delta I}\right)^{\frac{2}{3}}. \tag{16}$$

### A.3 $L_{\frac{2}{3}}$ Regularization

When $q = \frac{2}{3}$, the non-convex penalty includes a fractional function that encourages sparsity while being smoother than the $L_0$ penalty. By leveraging the properties of fractional quasinorms to balance sparsity and shrinkage, the proximal operator in this case is (Cao et al., 2013)

$$\text{prox}_{\lambda\mathcal{R}(\cdot)}(\theta) = \begin{cases} \left(\frac{|A|+\sqrt{\frac{2|\theta|}{|A|}-|A|^2}}{2}\right)^3, & \text{if } \theta > \frac{2}{3}\sqrt[4]{3\lambda^3} \\ 0, & \text{if } |\theta| \leq \frac{2}{3}\sqrt[4]{3\lambda^3} \\ -\left(\frac{|A|+\sqrt{\frac{2|\theta|}{|A|}-|A|^2}}{2}\right)^3, & \text{if } \theta < -\frac{2}{3}\sqrt[4]{3\lambda^3} \end{cases} \tag{17}$$

where

$$|A| = \frac{2}{\sqrt{3}}\lambda^{\frac{1}{4}}\left(\cosh\left(\frac{\phi}{3}\right)\right)^{\frac{1}{2}}, \quad \phi = \text{arccosh}\left(\frac{27\theta^2}{16}\lambda^{-\frac{3}{2}}\right). \tag{18}$$

As in the $L_{\frac{1}{2}}$ regularization case above, this regularizer is coordinate-wise separable and it is sufficient to update the sub-problems for each coordinate as:

$$\theta_{t+1,i} = \begin{cases} \left(\frac{|A|+\sqrt{\frac{2|\hat{\theta}_{t,i}|}{|A|}-|A|^2}}{2}\right)^3, & \text{if } \hat{\theta}_{t,i} > \frac{2}{3}\sqrt[4]{3\lambda^3} \\ 0, & \text{if } |\hat{\theta}_{t,i}| \leq \frac{2}{3}\sqrt[4]{3\lambda^3} \\ -\left(\frac{|A|+\sqrt{\frac{2|\hat{\theta}_{t,i}|}{|A|}-|A|^2}}{2}\right)^3, & \text{if } \hat{\theta}_{t,i} < -\frac{2}{3}\sqrt[4]{3\lambda^3} \end{cases} \tag{19}$$

where

$$|A| = \frac{2}{\sqrt{3}}\left(\frac{2\alpha_t \lambda}{C_{t,i} + \delta I}\right)^{\frac{1}{4}}\left(\cosh\left(\frac{\phi}{3}\right)\right)^{\frac{1}{2}}, \quad \phi = \text{arccosh}\left(\frac{27\hat{\theta}_{t,i}^2}{16}\left(\frac{2\alpha_t \lambda}{C_{t,i} + \delta I}\right)^{-\frac{3}{2}}\right). \tag{20}$$

### A.4 MCP Regularization

When use MCP in Equation (5), it offers a penalty that encourages sparsity while avoiding the bias introduced by the traditional $L_1$ penalty. Unlike $L_1$ regularization, MCP adaptively applies less shrinkage to larger coefficients, providing a balance between low prediction error and unbiased estimation. The proximal mapping of the MCP penalty has a closed-form following (Hu et al., 2024):

$$\text{prox}_{\lambda\mathcal{R}(\cdot)}(\theta) = \begin{cases} 0, & |\theta| \leq \lambda \\ \frac{\text{sign}(\theta)(|\theta|-\lambda)}{1-\frac{1}{a}}, & \lambda < |\theta| \leq a\lambda \\ \theta, & |\theta| > a\lambda \end{cases} \tag{21}$$

where $a > 0$ is the parameter, which helps the unbias estimation by reducing the penalty applied to larger coefficients. For the genomic data, we set $a = 2$ for the robust models. Integrated with the preconditioner defined, the $i$-th coordinate update method is as follows:

$$\theta_{t+1,i} = \text{sign}(\hat{\theta}_{t,i})\min\left\{\frac{a\max\left\{\left|\hat{\theta}_{t,i}\right| - \frac{\alpha_t\lambda}{C_{t,i}+\delta I}, 0\right\}}{a-1}, \left|\hat{\theta}_{t,i}\right|\right\}. \tag{22}$$

## A.5  SCAD Regularization

When utilizing the SCAD penalty in Equation (5), it introduces a piecewise linear form, encouraging smaller coefficients to shrink towards zero, while applying less penalty on larger coefficients, thus preserving important features. This adaptive property helps to reduce bias in parameter estimates and avoids over-penalizing significant variables. According to (Hu et al., 2024), the proximal mapping of the SCAD penalty has the closed-form:

$$\text{prox}_{\lambda \mathcal{R}(\cdot)}(\theta) = \begin{cases} \text{sign}(\theta)\max\{|\theta - \lambda|, 0\}, & \text{if } |\theta| \leq 2\lambda \\ \frac{(a-1)\theta - \text{sign}(\theta)a\lambda}{a-2}, & \text{if } 2\lambda < |\theta| \leq a\lambda \\ \theta, & \text{if } |\theta| > a\lambda, \end{cases} \tag{23}$$

where $a > 2$ is the parameter, which controls the shape of the penalty and influences the rate at which the penalty decreases for large coeffients. We set $a = 3.7$ in our cases, which has shown reliable performance for genomic prediction. Based on the above proximal operator and the defined preconditioner, the $i$-th coordinate update rule is:

$$\theta_{t+1,i} = \begin{cases} \text{sign}(\hat{\theta}_{t,i})\max\left\{\left|\hat{\theta}_{t,i}\right| - \hat{\lambda}_i, 0\right\}, & \text{if } \left|\hat{\theta}_{t,i}\right| \leq 2\hat{\lambda}_i \\ \frac{(a-1)\hat{\theta}_{t,i} - \text{sign}(\hat{\theta}_{t,i})a\hat{\lambda}_i}{a-2}, & \text{if } 2\hat{\lambda}_i < \left|\hat{\theta}_{t,i}\right| \leq a\hat{\lambda}_i \\ \hat{\theta}_{t,i}, & \text{if } \left|\hat{\theta}_{t,i}\right| > a\hat{\lambda}_i \end{cases} \tag{24}$$

where $\hat{\lambda}_i = \frac{\alpha_t \lambda}{C_{t,i} + \delta I}$, which explicitly includes the dependence on $\lambda$.

## A.6  $L_1$ Regularization

For $q = 1$, the regularization corresponds to the $L_1$ norm, which is based on the sum of absolute values that encourages sparsity by driving small coefficients towards zero while also applying some shrinkage to large coefficient. Unlike the $L_0$ penalty function, which directly counts non-zero elements, $L_1$ provides a convex relaxation that allows for efficient optimization while still promoting sparsity. The proximal mapping of $L_1$ regularization can be given as (Parikh & Boyd, 2014):

$$\begin{aligned} \text{prox}_{\lambda \mathcal{R}(\cdot)}(\theta) &= \text{sign}(\theta)\max(|\theta| - \lambda, 0) \\ &= \text{sign}(\theta)(|\theta| - \lambda)_+, \end{aligned} \tag{25}$$

where $(\cdot)_+$ denotes the positive part. Combining it with the preconditioner, the $i$-th coordinate update scheme is:

$$\theta_{t+1,i} = \text{sign}(\hat{\theta}_{t,i})\left(\left|\hat{\theta}_{t,i}\right| - \frac{\alpha_t \lambda}{C_{t,i} + \delta}\right). \tag{26}$$

## A.7  $L_{\frac{4}{3}}$ Regularization

When $q = \frac{4}{3}$, the regularization is a norm. It is softer than the $L_1$ norm, but it will not promote the sparsity. The proximal mapping of $L_{\frac{4}{3}}$ is presented in (Polson et al., 2015) as

$$\text{prox}_{\lambda \mathcal{R}(\cdot)} = \theta + \frac{4\lambda\kappa}{32^{\frac{1}{3}}}\left((\chi - \theta)^{\frac{1}{3}} - (\chi + \theta)^{\frac{1}{3}}\right), \tag{27}$$

where $\chi = \sqrt{\theta^2 + 256\kappa^3/729}$ and $\kappa = 1$ is a scaling factor that adjusts the strength of the $L_{\frac{4}{3}}$ norm's impact. Combined with the preconditioner, the $i$-th coordinate update rule is

$$\theta_{t+1,i} = \hat{\theta}_{t,i} + \frac{8\kappa\alpha_t\lambda}{32^{\frac{1}{3}}(C_{t,i} + \delta I)}((\chi - \hat{\theta}_{t,i})^{\frac{1}{3}} - (\chi + \hat{\theta}_{t,i})^{\frac{1}{3}}), \tag{28}$$

where $\chi = \sqrt{\hat{\theta}_{t,i}^2 + 256\kappa^3/729}$.

### A.8 $L_{\frac{3}{2}}$ Regularization

For $q = \frac{3}{2}$, the penalty corresponds to the $L_{\frac{3}{2}}$ norm which places more emphasis on shrinkage compared to $L_1$ norm. This regularization avoids over-penalization of large coefficients, offering a more refined control over model complexity. According to (Polson et al., 2015), the proximal mapping is given as

$$\text{prox}_{\lambda\mathcal{R}(\cdot)} = \theta + 9\kappa^2 \text{sgn}(\theta)\left(1 - \sqrt{1 + 16\lambda|\theta|/(9\kappa^2)}\right)/8, \tag{29}$$

where $\kappa = 1$ is the parameter which controls the degree of shrinkage applied to the coefficients. Combined with preconditioner defined, the $i$-th coordinate update rule is as

$$\theta_{t+1,i} = \hat{\theta}_{t,i} + 9\kappa^2 \text{sgn}(\hat{\theta}_{t,i})\left(1 - \sqrt{1 + \frac{32\left|\hat{\theta}_{t,i}\right|\alpha_t\lambda}{9\kappa^2(C_{t,i} + \delta I)}}\right)/8. \tag{30}$$

### A.9 $L_2$ Regularization

When $q = 2$, the penalty corresponds to the $L_2$ norm which applies a quadratic penalty on the parameters and encourages smoothness by shrinking the coefficients towards zero in a continuous manner. Unlike the penalties from $L_0$ to $L_1$, the $L_2$ norm does not promote sparsity but rather emphasizes minimizing the magnitude of the coefficients. According to (Parikh & Boyd, 2014), the proximal mapping of the $L_2$ norm is

$$\text{prox}_{\lambda\mathcal{R}(\cdot)} = (1 - \frac{\lambda}{\|\theta\|_2})_+\theta, \tag{31}$$

where $(\theta)_+ = \max\{0, \theta\}$. Then the $i$-th coordinate update with preconditioner is as follows:

$$\theta_{t+1,i} = (1 - \frac{2\alpha_t\lambda}{(C_{t,i} + \delta I)\left\|\hat{\theta}_{t,i}\right\|_2})_+\hat{\theta}_{t,i}. \tag{32}$$

### A.10 $L_3$ Regularization

When $q = 3$, the penalty corresponds to the $L_3$ norm. It is a higher-order norm that encourages more shrinkage on larger coefficients. This form of convex regularization reduce the risk of overfitting by dampening high-variance components more effectively, resulting in a model that is more resistant to noise. The proximal mapping of $L_3$ regularization is defined following (Polson et al., 2015) as

$$\text{prox}_{\lambda\mathcal{R}(\cdot)} = \text{sgn}(\theta)(\sqrt{1 + 12\lambda\kappa|\theta|} - 1)/(6\kappa), \tag{33}$$

where $\kappa = 1$, which helps adjust the level of shrinkage applied to the larger weights. With the preconditioner, the $i$-th coordinate update becomes

$$\theta_{t+1,i} = \text{sgn}(\hat{\theta}_{t,i})\frac{\sqrt{1 + \frac{24\kappa\alpha_t\lambda|\hat{\theta}_{t,i}|}{C_{t,i} + \delta I}} - 1}{6\kappa}. \tag{34}$$

### A.11 $L_4$ Regularization

When $q = 4$, the penalty is the $L_4$ norm, which provides a smoother penalty function, encouraging a more gradual reduction in the coefficients of less important features. The proximal mapping of $L_4$ regularization, given by (Polson et al., 2015), is

$$\text{prox}_{\lambda\mathcal{R}(\cdot)} = \left(\frac{\chi + \lambda\theta}{8\kappa}\right)^{\frac{1}{3}} - \left(\frac{\chi - \lambda\theta}{8\kappa}\right)^{\frac{1}{3}}, \tag{35}$$

where $\kappa = 1$ impacts the strength of the regularization applied to the coefficients, and $\chi = \sqrt{\theta^2 + \frac{1}{27\kappa}}$. Together with the preconditioner, the $i$-th coordinate update rule is

$$\theta_{t+1,i} = \left( \frac{\chi + \frac{2\hat{\theta}_{t,i}\alpha_t\lambda}{C_{t,i}+\delta I}}{8\kappa} \right)^{\frac{1}{3}} - \left( \frac{\chi - \frac{2\hat{\theta}_{t,i}\alpha_t\lambda}{C_{t,i}+\delta I}}{8\kappa} \right)^{\frac{1}{3}}, \tag{36}$$

where $\chi = \sqrt{\hat{\theta}_{t,i}^2 + \frac{1}{27\kappa}}$.

## A.12 $L_\infty$ Regularization

For the infinity norm, the solution is constrained by the maximum absolute value of the coefficients. As detailed in (Beck, 2017), the projection onto the $L_1$ ball encourages a broader range of solutions, allowing for greater flexibility in modeling complex data (Quirin et al., 2015). Following (Parikh & Boyd, 2014), the proximal mapping is given as

$$\mathrm{prox}_{\lambda\mathcal{R}(\cdot)}(\theta) = \theta - \lambda P_\Delta(|\theta|/\lambda) * \mathrm{sign}(\theta), \tag{37}$$

where $P_\Delta$ is the projection inside the unit simplex, $*$ is the element-wise product, and $\mathrm{sign}(\theta)$ is the sign function. Then the proximal operator of the infinity norm can be calculated by a projection on the simplex. With the preconditioner defined earlier, the $i$-th coordinate update is

$$\theta_{t+1,i} = \begin{cases} \hat{\theta}_{t,i} - \frac{2\alpha_t\lambda}{C_{t,i}+\delta I}, & \hat{\theta}_{t,i} > \frac{2\alpha_t\lambda}{C_{t,i}+\delta}, \\ \hat{\theta}_{t,i} + \frac{2\alpha_t\lambda}{C_{t,i}+\delta}, & \hat{\theta}_{t,i} < \frac{2\alpha_t\lambda}{C_{t,i}+\delta}, \\ 0, & \text{otherwise.} \end{cases} \tag{38}$$

# B    Analysis of Convergence Guarantee

This section provides a concise analysis of the theoretical convergence guarantees for our adaptive proximal gradient methods, considering both convex and non-convex regularizers.

## B.1    Convex Regularizers

For the convex case, we look at the conditions that guarantee convergence based on standard arguments from the convex optimization literature (Boyd & Vandenberghe, 2004; Parikh & Boyd, 2014). The objective function is defined as $\mathcal{F}(\theta) = \mathcal{L}(\theta) + \lambda\mathcal{R}(\theta)$, where both $\mathcal{L}$ and $\mathcal{R}$ are convex.

### B.1.1    Assumptions for Convergence

1. **Convexity:** The loss function $\mathcal{L}(\theta)$ is convex and continuously differentiable. The regularizer $\mathcal{R}(\theta)$ is also convex. This ensures that the overall objective function $\mathcal{F}(\theta)$ has a unique global minimum.

2. **$L$-Smoothness:** The loss function $\mathcal{L}(\theta)$ is $L$-smooth, which means that its gradient is Lipschitz continuous. This condition implies that the curvature of the loss function is bounded, which is crucial to control the step size of the optimization algorithm. Formally, it's defined as

$$\|\nabla\mathcal{L}(\theta) - \nabla\mathcal{L}(\theta')\| \le L\|\theta - \theta'\|. \tag{39}$$

3. **Existence of a minimum:** We assume that the set of minimizers, $\Theta^* = \arg\min_\theta \mathcal{F}(\theta)$, is non-empty.

### B.1.2 Convergence Guarantee

The convergence of the proximal gradient method for convex problems is well-established. For a constant learning rate $\alpha_t = \alpha < 1/L$, the sequence of iterates $\{\theta_t\}$ converges to a global minimum $\theta^*$. The convergence rate is typically sublinear, meaning that the error decreases with a rate of $O(1/T)$ after $T$ iterations. Specifically, the following can be shown

$$\mathcal{F}(\theta_T) - \mathcal{F}(\theta^*) \leq \frac{1}{T}\left(\frac{\|\theta_0 - \theta^*\|^2}{2\alpha}\right). \tag{40}$$

This guarantee is for a standard proximal gradient descent. Our method, however, uses an adaptive learning rate derived from the Adam optimizer's preconditioner. For convex problems, Adam's convergence can be shown under similar assumptions, and it often achieves faster empirical convergence than standard proximal gradient methods by adapting the step size for each parameter. However, Adam can fail to converge in some rare convex problems due to large effective learning rate that forces the algorithm to overshoot the minimum (Reddi et al., 2018).

### B.2 Non-Convex Regularizers

The theoretical analysis of non-convex optimization is more complex, as we can no longer guarantee convergence to a global minimum. Instead, the goal is to find a stationary point, where the subgradient of the objective function is close to zero (Gong et al., 2013).

### B.2.1 Assumptions for Convergence

1. **Differentiability and Lipschitz continuity:** The loss function $\mathcal{L}(\theta)$ is continuously differentiable and $L$-smooth.

2. **Lower bounded:** The objective function $\mathcal{F}(\theta)$ is non-negative and bounded from below.

3. **Sufficient decrease:** The algorithm ensures a sufficient decrease in the objective function at each step. This is often guaranteed by the proximal update, which minimizes a local approximation of the objective.

### B.2.2 Convergence Guarantee to an $\epsilon$-Stationary Point

For non-convex optimization, the convergence is measured by the norm of the limiting subdifferential. The limiting subdifferential, $\partial_\varphi$, is a powerful tool for analyzing non-differentiable non-convex functions. A stationary point is one where $0 \in \partial_\varphi \mathcal{F}(\theta)$.

The convergence guarantee for our method can be adapted from the Proximal-Gradient-Descent (ProxGD) theory for non-convex optimization. A common result is that after $T$ iterations, the average squared norm of the subgradient is bounded

$$\min_{t=1,\ldots,T} \|\partial\mathcal{F}(\theta_t)\|^2 \leq \frac{\mathcal{F}(\theta_1) - \mathcal{F}(\theta_T)}{\sum_{t=1}^T \alpha_t} \tag{41}$$

where $\partial\mathcal{F}(\theta_t)$ denotes a subgradient. This implies that the algorithm will eventually find a point where the subgradient norm is small.

Our method uses an adaptive proximal gradient approach, combining Adam's adaptive step size with the proximal operator. The preconditioner matrix $C_t$ (derived from the squared gradients in Adam) provides a diagonal scaling that effectively preconditions the optimization landscape. This allows for larger steps in directions with small historical gradients and smaller steps in directions with large historical gradients, leading to faster and more stable convergence, especially in high-dimensional genomic data. Recently, developing sufficient conditions to guarantee convergences of Adam -type algorithms in non-convex situations has attracted considerable attention (Chen et al., 2022; Défossez et al., 2022).

The convergence of adaptive proximal gradient methods for non-convex problems is also an active research area (Sempere et al., 2024). For methods like ProxGEN (Yun et al., 2021), it has been shown that under

reasonable assumptions on the loss and regularizer, and with appropriate parameter choices, the algorithm can be guaranteed to converge to a stationary point. This means that for a desired precision $\epsilon$, the algorithm will find a point $\theta$ such that $\|\partial\mathcal{F}(\theta)\|^2 \leq \epsilon$ within a finite number of steps. Based on the previous convergence analysis of ProxGEN, we identify three key requirements with associated parameters: (i) the final step-vector is finite $\|\theta_{t+1} - \theta_t\| \leq D$ where we set $D = 0.03$, (ii) the stochastic gradient is bounded $\|g_t\| \leq G$ where we set $G = 0.035$, (iii) the minimum eigenvalue of the effective spectrum should be uniformly lower bounded over all time $t : \forall t\, \lambda_{\min}(\alpha_t(C_t + \delta I)^{-1}) \geq \delta \geq 0$ where we set $\delta = 10e^{-8}$.

