# OpenReview forum: "Proximal Regularization of Deep Residual Neural Networks Applied to High-dimensional Genomic Data"
_TMLR — Rejected by TMLR_

### Review · Reviewer_6Zg9 · 2025-08-14

**Summary Of Contributions:**

Contributions of the paper are:

1. Develop stochastic adaptive proximal gradient methods for ResNets incorporating both convex and non-convex regularizers (from $L_0$ to $L_{\infty}$)

2. Extensive evaluation on real high-dimensional genomic dataset. Compare different network depths and regularizers to find optimal configurations.

**Audience:**

Yes

**Audience Explanation:**

- $\textbf{Methodological novelty}$: The integration of Adam-style preconditioning into a proximal-gradient framework that works with both convex and non-convex regularizers is an algorithmic contribution, not just an application tweak.


- $\textbf{Empirical evaluation in a challenging regime}$: The focus on high-dimensional, small-sample data (genomics) is a well-known hard setting for deep learning, so showing that deep ResNets can beat strong sparse linear baselines is notable.


- $\textbf{Breadth of regularizers}$: Sweeping from $L_0$ to $L_\infty$ and including MCP/SCAD gives the results methodological breadth that might appeal to optimization and statistics-minded ML researchers.

**Broader Impact Concerns:**

- The method works on animal/plant genetics here, but could be applied to human genomic data. That raises risks around privacy (leaking genetic traits) and discrimination (misuse in insurance, hiring, etc.)

**Claims And Evidence:**

Yes

**Claims Explanation:**

$\textbf{Claim}$: Adaptive proximal gradient ResNets outperform traditional sparse proximal methods.


$\textbf{Evidance}$:
- They run side-by-side experiments on three independent real genomic datasets, across many regularizers, and consistently show lower test MSE for ResNets (especially with $L_{\frac{1}{2}}$) than for sparse linear models.


$\textbf{Claim}$: Proximal regularization is effective for high-dimensional noisy data.


$\textbf{Evidance}$:
- ResNet + proximal regularizers outperform both unregularized and traditionally regularized setups in their benchmarks.
- **However results are limited to genomic prediction; generality to other domains isn’t tested.**

**Requested Changes:**

All experiments are on genomics-like small-n, high-p regression

- How does the method behave on standard ML benchmarks with different statistical properties (e.g., CIFAR-10 tabularized, UCI datasets)?


- Is the observed advantage of $L_{\frac{1}{2}}$ specific to the correlation structure of SNP data?

---

> ### Author Response · Authors · 2025-09-18
> **Response to reviewer comments**
>
> We sincerely appreciate you taking the time to review our manuscript and for providing insightful and constructive feedback. The method suggested in this study was specifically designed for the small-$n$, high-$p$ setting, which is a characteristic of genomic prediction, where the number of features (genomic markers) is orders of magnitude larger than the number of samples. Standard ML benchmarks such as CIFAR-10 or most UCI datasets are fundamentally different: they typically fall into the large-$n$, moderate-$p$ regime, where data richness allows standard deep learning and gradient-boosted tree methods to perform very strongly.
>
> For completeness, we performed experiments on the CIFAR-10 dataset. However, we did not get competitive results with our method, which is consistent with the fact that our approach is not designed for that type of data. This result highlights a crucial point: the strength of the proposed approach lies in its ability to leverage adaptive proximal methods with ResNets to tackle the unique statistical challenges of high dimensionality, irregular feature correlation patterns, and limited sample sizes that are common in genomic datasets. Hence, we did not report the results on the CIFAR-10 dataset.
>
> The $L_{\frac{1}{2}}$ regularizer is a non-convex penalty that strikes a unique balance between the aggressive feature selection of the $L_0$ norm and the continuous shrinkage of the $L_1$ norm. Unlike the $L_1$ norm, which tends to arbitrarily select only one feature from a group of highly correlated features and shrink the rest to zero, the $L_{\frac{1}{2}}$ norm is better suited to handle these correlated blocks by distributing the feature importance more equitably. This property allows our model to retain valuable information from these correlated groups, leading to a more robust and accurate prediction.

---

### Review · Reviewer_bawG · 2025-08-19

**Summary Of Contributions:**

This paper studies regularization techniques for training Resnets for genomic prediction tasks (which present specific challenges due to high-dimensional noisy data).
The main contribution of the paper is an empirical evaluation on various Resnets architectures combined with different regularization penalties for the training loss (convex, nonconvex, nonsmoooth), trained via a proximal ADAM-like algorithm.
The novelty of the paper lies mainly in the experimental study: the paper applies the existing ProxGen [1] algorithm to genomics.

[1] Adaptive Proximal Gradient Methods for Structured Neural Networks Yun, Lozano, Yang


**Strengths**

The paper brings a clear focus on regression tasks for genomic data, which seems to remain a challenging task for deep learning methods, and provides a quantitative benchmark with the aim of offering practical guidelines.

**Weaknesses**



First, and it is not a weakness in itself, the method used is not new and is presented in a very similar manner in [1], where some experiments are already done on ResNets for images using nonconvex penalties ($L_{1/2}, SCAD, MCP$). Much of the technical content (including closed-form updates for various penalties in the appendix) comes from that work. In that extend, I find the abstract misleading: the sentence "In our study, we develop a stochastic adaptive proximal gradient ResNet method that can handle" suggests the paper develops a new method.
Among the two stated contributions: 1. developping stochastic adaptive proximal gradient methods for ResNets and 2. providing a comprehensive benchmark evaluation of networks size and regularizations, I believe only the second item is novel and constitutes the real contribution.

Given that the method is not new, the novelty and interest of the paper essentially lies in the experimental content.
 Yet, on that side, I think that the experiments section is not sufficiently clear nor convincing:
- the presentation of the results could be enhanced (for example, avoiding quantitative results put directly in the core of the text, adding plots)
- some more insights could be given on the interest of such nonconvex nonsmooth penalties (what is final sparsity in the network ? do some penalties enable faster convergence (plot convergence curves) ? etc)
- benchmarking against more baselines (Resnet without regularization, alternative architectures,)

Beyond these remarks,  there are some other confusing points that I detail below.


**Questions**


*Introduction*

The main motivation of the paper is stated page 2 "However, when dealing with genomic data, the challenge of fitting ResNets becomes even more pronounced, not only because of the high dimensionality but also because of the inherent noise within the data.": this deserves to be better supported by some empirical examples or references. More generally, could the authors motivate more why ResNets would be more relevant than other (sparse or not) neural networks for genomics data ?

*Section 2, Related Work*

The paper does not clearly distinguish between 1. the choice of regularization penalty in the loss ($L_{1/2}, L_1, etc$ and 2. the optimization algorithm (proximal methods, SGD, subgradient methods).
In the related works and later on, the distinction is not really made as these are both encompassed under the term "proximal regularization". So I think a clearer separation would avoid confusion and allow the authors to better situate related works on both aspects.

*Section 3 Methods*

- Equation 4, what does the exponent in prox means $\mathrm{prox}^{C_t + \delta \mathrm{I}}$, is this related to a preconditioned norm?
- Why is $C_t = \left( \beta C_{t-1} + (1-\beta) g_t^2 \right)^{\frac 1 2}$ and not $C_t = \left( \beta C_{t-1}^2 + (1-\beta) g_t^2 \right)^{\frac 1 2}$
(This follows the original paper, but then  thederivation in their appendix seems inconsistent.)
- The remarks in Sec. 3.3 (on convergence theory) appear to be largely taken from the initial paper without clear added value (for example, Advantages of using adaptive gradient methods in Theorem 1).  Maybe the initial goal of the authors was to make connections between the assumptions required for the theory to hold and the specific regression task on genomics, but there is no clear insight in this sense. For instance, in the first remark "On convergence results", the authors write that the constants D and G should be small but as far as I understand, the authors have no control on these values. I guess it could at least be experimentally checked.
- The paper writes "We extend the application of sparse $L_q$ regularization beyond the conventional range $0 \leq q \leq 1$. These are not sparsifying and could be optimized directly with ADAM, making proximal methods unnecessary right ?

*Section 4 Implementation details*

In my opinion, this section could be rewritten more clearly:
1. Stating the prediction task (regression).
2. Specifying the regularization penalties tested (convex vs. nonconvex, smooth vs non smooth).
3. Describing the models used for benchmark (linear regression, ResNets of various depths).

*Section 5 Experiments*

- As the paper is mostly experimental, the setting could be explained in more details: what is the loss $\mathcal L$ used ? Does it satisfy the assumptions of the theoretical framework?
- 5.3 Adaptive Proximal Methods vs Traditional Sparse Proximal Methods: I don't understand this terminology. I believe the authors compare 1. deep-based methods + nonconvex regularisation with 2. linear regression + non convex regularisation. It looks like the term adaptive is chosen all over the paper because of the adaptive optimization algorithm used to train the neural network but as far as I understand, the main difference is deep vs linear methods.
- As stated earlier, results presentation could be improved with plots (e.g., performance vs. ResNet depth).

**Additional Comments:**

None

**Audience:**

No

**Audience Explanation:**

The paper could interest part of the TMLR audience as the goal is to provide guidelines to train Resnets for genomic regression but the experiments are relatively limited, and do not giving enough insights which reduce the paper's interest for the community.

**Claims And Evidence:**

Yes

**Claims Explanation:**

The main takeaways of the paper are experimental and appear accurate,
- on regularization, nonconvex nonsmooth penalties can improve performance (which is aligned with the results of [1] on ResNets on images)
- on the architecture size,  increasing depth helps which is not surprising.

However, the experiments lack comparison with other baselines and deeper analysis of the results, making the paper not particularly convincing (see weaknesses).

**Requested Changes:**

1. Clarify contributions: to which extend the authors adapt ProxGen for ResNets ?
2. Provide stronger empirical or literature support to motivate the need for specific new regularization techniques to train ResNets on genomics.
3. Clarify terminology (distinction between regularization penalties and optimization methods)
4. Improve experiments (presentation, baselines, deeper analysis of the resulting models)

---

> ### Author Response · Authors · 2025-09-18
> **Response to reviewer comments**
>
> Thank you so much for reviewing our manuscript and for providing valuable comments. The depth of the network is not just about increasing its capacity; it's about enabling it to discover the multi-level, complex patterns that are fundamental to do predictions. In our study, ResNets were utilized on high-dimensional genomic data for prediction tasks because they can overcome the vanishing gradient problem. By enabling the use of a deep architecture, ResNets facilitate the learning of hierarchical features - from individual genetic variants to more complex, higher-level genomic patterns - that are essential for robust prediction tasks on the high-dimensional genomic data.
>
> In Subsection 3.2, the exponent in Equation (4) is not a standard exponent but a superscript notation for a preconditioned proximal operator. This is a way in the optimization literature to denote a proximal operator computed with respect to a non-identity inner product, which is defined by the preconditioning matrix $C_t + \delta I$. This matrix modifies the standard Euclidean norm to accelerate convergence, a key feature of adaptive gradient methods. For methods like ProxGEN, it has been shown that under reasonable assumptions on the loss and regularizer, and with appropriate parameter choices, the algorithm can be guaranteed to converge to a stationary point. This means that for a desired precision $\epsilon > 0$, the algorithm will find an optimal point $\mathbf{x}$ within a finite number of steps. Based on the previous convergence analysis of ProxGEN, we identify three key requirements with associated parameters: (i) the final step-vector is finite $D$, where we set $D=0.03$, (ii) the stochastic gradient is bounded $G$, where we set $G=0.035$, (iii) the minimum eigenvalue of the effective spectrum should be uniformly lower bounded over all time, where we set $P_t = \gamma + \alpha_t \delta I \ge \epsilon > 0$. These details have been introduced in the last paragraph of Appendix B. At the same time, we simplified the convergence analysis for algorithm 1 in Appendix B. For $L_q$ with $q > 1$, the regularization penalty is differentiable and could be optimized directly with a standard Adam optimizer. Because we are doing regression, we have used $L_2$ as a loss $\mathcal{L}$. It satisfies the assumptions of the theoretical framework very well. Below follows replies to Requested Changes.
>
> Reply: In the revised version, we clarify the contributions in the manuscript in the last part of the Introduction (In Section 1).
>
> Reply: We revised Related Work (Section 2) to highlight recent studies showing the benefits of nonconvex penalties in high-dimensional biology. These works collectively motivate the exploration of $L_{\frac{1}{2}}$, MCP, SCAD, and other nonconvex penalties in genomic deep learning, where sparsity, handling of correlated predictors, and robustness against over-fitting are critical. We also strengthened the Introduction (Section 1) to better explain why conventional $L_1$ or $L_2$ penalties are insufficient for genomic prediction and why adaptive proximal methods can yield a better generalization.
>
> Reply: We have reorganized Related Work (Section 2) to clarify this terminology. We have explicitly distinguished between (1) regularization penalties (e.g., $L_1$, $L_{\frac{1}{2}}$, MCP, SCAD), which impose structural constraints or sparsity on parameters, and (2) optimization methods (e.g., proximal gradient descent, Adam, ProxGen), which determine how parameters are updated during training.
>
> Reply: In the revised version, we (1) have improved the presentation of training loss curves (Figure 3 in the third part of Section 5), (2) add LightGBM with adequate tuning as a new baseline method for more extensive performance evaluation (in the second part of Section 5), (3) include a deeper analysis of the models by reporting feature sparsity patterns, distance correlation (dCor), and computing efficiency across data sets (Table 1-4 in Section 6), and (4) report the sparsity analysis of ResNets on the datasets (Table 5 in Subsection 6.3) and give relative analysis in Subsection 6.3. These additions will provide stronger empirical validation and make the experimental section more complete and transparent. The relative comparison and analyses have been discussed in Section 6.

---

### Review · Reviewer_rYgN · 2025-08-22

**Summary Of Contributions:**

The paper presents empirical results on the ProxGEN method on genomic data.
ProxGEN is a method for regularizing deep residual neural networks by
introducing a proximal term in the loss function. The method is compared in
terms of out-of-sample prediction performance against a regularized linear
regression method and a ProxGEN ResNet with no regularization. The authors
report that ProxGEN outperforms the other methods in terms of prediction
performance on the genomic data sets.

## Strengths

- The background is extensive and the description of the ProxGEN method is clear
  and well-structured.
- The problem is important and relevant.
- The writing is generally clear and easy to follow.

## Weaknesses

- The main weakness is that the paper offers little new insight. The ProxGEN method has
  already been introduced in Yun et al. 2020 and the current paper, as I
  understand it, really only offers an application to genetic data and investigation of
  the various types of segularization. I am not
  sure if this is enough to motivate publication in TMLR, but I view this as a
  marginal addition to the literature. To me it seems to perhaps fall into the category of a papers that "re-implent an idea that has already been produced before".
  Most of the paper is made up of background and description of the ProxGEN method. The actual new, empirical, results make up
  much less, and are also generally not as well-presented.

- Anonymity is violated since the GitHub repo linked is not anonymous.

- The authors list several other techniques to combat overfitting (e.g. early
  stopping and drop-out), but only compares their method against ResNets with no
  regularization and a regularized linear regression method.

- The numerical performance of the method is not investigated thoroughly. There
  is a complexity analysis (without derivation), but no empirical results.

- The presentation of the results in the paper is not very well-developed. All
  the results are presented either inline or in tables, which are quite hard to
  grasp.

- The code in the software repository is not well-structured. It concists of several Jupyter notebooks, without much explanation regarding which of the experiments these correspond to. The instructions are missing clear steps
  for reproducing the results, including dependencies for the code. Comments in
  the code are also in Chinese (I think), which makes it hard to follow. Paths
  to data sets in the code are also hard-coded to `/root/`, which is not exactly
  portable. The raw results that I can see in the notebooks should be saved to
  some files are not uploaded to the repository. It' also confusing that the
  paper mentions ResNet5 and ResNet25, but in the code I only see ResNet18, 34,
  50, 101, and 152. These comparisons between the complexity should also be
  presented more clearly in the paper. It's hard to get an overview when all the
  results are just inlined in the text.

- To me, one of the most interesting aspects of applying regularization methods
  the ability to generate sparse networks. The authors, however, do not
  investigate this at all.

- The background in the paper, while extensive, does not really provide an
  overview of research that specifically concerns the application of machine
  learning methods to prediction taks in genomics. Just searching the web for
  "genomic prediction machine learning" yields a number of papers that seems to
  be relevant not just for the background, but also for the comparison of
  methods. Here are just a few that I found that seem to be relevant. (I have not
  looked at these in detail.)
  1. Wu H, Gao B, Zhang R, Huang Z, Yin Z, Hu X, Yang CX, Du ZQ. Residual
     network improves the prediction accuracy of genomic selection. Anim Genet.
     2024 Aug;55(4):599-611. doi: 10.1111/age.13445. Epub 2024 May 15.
     PMID: 38746973.
  2. Lourenço, V., Ogutu, J., Rodrigues, R. et al. Genomic prediction using
     machine learning: a comparison of the performance of regularized
     regression, ensemble, instance-based and deep learning methods on synthetic
     and empirical data. BMC Genomics 25, 152 (2024).
     <https://doi.org/10.1186/s12864-023-09933-x>
  3. Montesinos-López OA, Montesinos-López A, Pérez-Rodríguez P, Barrón-López
     JA, Martini JWR, Fajardo-Flores SB, Gaytan-Lugo LS, Santana-Mancilla PC,
     Crossa J. A review of deep learning applications for genomic selection. BMC
     Genomics. 2021 Jan 6;22(1):19. doi: 10.1186/s12864-020-07319-x. PMID:
     33407114; PMCID: PMC7789712.
  4. Fan, Y., Waldmann, P. Multi-task genomic prediction using gated residual
     variable selection neural networks. BMC Bioinformatics 26, 167 (2025).
     <https://doi.org/10.1186/s12859-025-06188-z>

**Additional Comments:**

Here are some minor remarks regarding the paper.

### Abstract

You say that you "develop a stochastic adaptive proximal gradient ResNet
method", but that's not actually new, right? The method was already introduced
in Yun et al. 2020. Be more precise about what your contributions are.

### Introduction

- Paragraph 3, last sentence. You say the regularization effect is too weak, but
  you don't explain why or give any reference.

### Related Work

- Paragraph 3: You mention normalization methods here, but it's not actually
  clear to me how they related to the problem at hand, which I understand to be
  overfitting.

### Methods

- Paragraph 1: Can Hinton 2012 really be a good reference for why ResNets (which
  were developed in 2015) also suffer from overfitting?
- Heading 3.2: Regularizer -> Regularizers
- Figure 1: The font sizes are inconsistent and too small, especially in 1d.

### Implementation Details

- Paragraph 1: The first sentence is awkward.

### Experimental Results

- Section 5.3, first sentence. Replace traditional sparse proximal method with a
  more descriptive term, such as regularized linear regression.

**Audience:**

Yes

**Audience Explanation:**

It's hard to answer "No" to this question, since I'm sure there are _some_
individuals in TMLR's audience that would be interested in the results here.
However, since the paper is mostly an application of an existing method, I think
the interest is limited and would perhaps be better suited for an outlet with a
more applied focus.

**Broader Impact Concerns:**

I don't see any major ethical concerns with the work presented in the paper.

**Claims And Evidence:**

No

**Claims Explanation:**

In line with my previous comments, I think the experiments would need to be
developed further. I would have liked to see comparisons against other methods
with which to combat overfitting in general. I also think that the literature
review should be more focused on methods in genomics, and seems to be to
currently be more about general methods for deep learning. Finally, the code is
also not well-structured, which makes it hard to investigate the results in
detail.

**Requested Changes:**

- Provide a link to an anonymized repository for the code, or just submit it along with your paper as a supplement.
- Provide a more thorough investigation of the computational performance of the
  method.
- I would like to see comparisons against other methods with which to combat
  overfitting in deep learning, such as early stopping and dropout as well as
  deep learning architectures specialized for genomic data.
- Investigate and discuss sparsity.
- Provide a more thorough literature review of methods for genomic prediction
  using machine learning.
- Organize the code in the repository better, with clear instructions for
  reproducing the results, comments that are readable to an internatinal
  audience, and publish raw data from the results of the experiments.
- Present the results in a more structured way, e.g. by using figures and
  separating the results from the text.
- It would be good to include a precise list of contributions somewhere in the
  paper, perhaps at the end of the introduction.
- There is a bit of overlap between the Introduction and the Related Work
  section. For instance, you explain dropout twice. Please have a second look at
  the structure of the paper and see if you can improve this.
- You mention normalization in your paper and use batch normalization. However,
  I don't see any discussion on the interplay between this type of normalization
  and regularization. It is well-known that this is important in the context of
  feature normalization. Does it also apply to the context of batch
  normalization?
- Did you normalize the features? And if so, how? This is at least very
  important for the regularized linear regression method, so would be important
  to mention.

---

> ### Author Response · Authors · 2025-09-18
> **Response to reviewer comments**
>
> Thank you so much for reviewing our manuscript and for providing valuable comments. Our work extends ProxGen in several important ways. First, we adapt the framework to deep residual networks (ResNets) for high-dimensional genomic data, which have not been considered in the original study. Second, we considerably broaden the class of penalties. Third, we provide convergence analysis tailored to this setting. These extensions go beyond a simple re-implementation and represent a substantive methodological and empirical contribution. We expanded the Related Work to include recent advances in machine learning for genomic prediction. The widely used LightGBM has been added as a strong nonlinear baseline method. New metrics such as distance correlation (dCor), sparsity, and time consumption have been analyzed and reported. Training loss cures and relative analysis have been added as well as complexity analysis for the methods.
>
> Below are replies to Requested Changes.
>
> Reply: We now provided a link to an anonymized Github repository for the code (in the last part of Section 5). All the codes are well-organized with a clear structure and comments to enhance readability and understanding. We also uploaded all raw data from the experiments to the repository to ensure full transparency and reproducibility.
>
> Reply: We now provided a more comprehensive comparison and analysis for the ResNets with different regularizers and the traditional methods on four datasets (In Section 6). New plots for the training losses on the mice, pig, wheat, and pine data with mainly ResNet-18 and ResNet-15 for their outstanding performance for the prediction tasks (In Section 5). We reported the sparsity comparison in Table 5 (Subsection 6.3). Relative analysis has been included in Subsection 6.3.
>
> Reply: Our study specifically focuses on adaptive proximal regularization of ResNets, and the main objective is to compare different regularizers within this framework rather than exhaustively benchmark all existing overfitting-control techniques. ResNets themselves already represent a state-of-the-art deep architecture for genomic data and combining them with proximal regularization provides a novel contribution beyond standard approaches such as early stopping or dropout. These techniques can certainly be applied in practice alongside our method, but to keep the scope clear we restricted the study to ResNets with varying regularizers and compared them against baseline methods (linear sparse proximal methods and LightGBM). We have clarified this design choice in the revision and expanded the Related Work (Section 2) to discuss complementary strategies like dropout and early stopping.
>
> Reply: We reported the sparsity comparison for ResNet-18 (mice and pig data) and ResNet-15 (wheat and pine data) for four genomic datasets (Table 5 in Subsection 6.3). Relative comparison and analysis have been included in Subsection 6.3.
>
> Reply: We significantly expanded Related Work (Section 2) to include recent studies in genomic prediction using deep learning and machine learning to provide stronger context for our contribution.
>
> Reply: The revised repository now includes a readme.txt file, step-by-step instructions, and English documentation. Hard-coded paths were removed, comments were translated, and all raw result files have been uploaded. The link now is in the last part of Section 5.
>
> Reply: We improved the presentation by including training loss curves (Figure 3 in Section 5), reorganizing results into structured tables, and separating results from the main text for clarity (Table 1-4 in Section 6).
>
> Reply: We added a dedicated list of contributions at the end of the Introduction (Section 1), clearly summarizing the novelty of our approach.
>
> Reply: We updated the Introduction (Section 1) and Related Work (Section 2) to remove redundancy (e.g., dropout is now explained only once). The structure is streamlined for better readability.
>
> Reply: We reorganized the Related Work (Section 2) to highlight the interplay between batch normalization and proximal regularization. While normalization stabilizes training and reduces internal covariate shift, it does not enforce sparsity. Proximal regularization complements batch normalization by explicitly controlling parameter magnitudes and sparsity, which is especially important in high-dimensional genomic data.
>
> Reply: Yes, all features were normalized to zero mean and unit variance prior to training. This preprocessing step was applied consistently across ResNet models, traditional sparse linear models, and LightGBM with adequate tuning (in Section 5).

---

### Decision · Action_Editor_vP7A · 2025-10-17

**Recommendation:** Reject

**Audience:**

No

**Audience Explanation:**

In its current form, the paper does not contain meaningful findings, as it presents an existing algorithm, and the experimental validation is not complete ; the provided code is hard to exploit in its current form.

**Claims And Evidence:**

No

**Claims Explanation:**

Both reviewers `bawG` and `rYgN`  raise serious concerns about the contribution.
First, the authors state ""In our study, we develop a stochastic adaptive proximal gradient ResNet method that can handle", but the ProxGEN algorithm is used as is and is not a contribution of the authors.
Second, since there is no new theory (only application of an existing algorithm, ProxGen, on genomic data), the paper should at least display extensive validation, which it does not.
The code contribution is also not high enough, as it consists in jupyter notebooks which are hard to exploit (as opposed to a proper modular python package)

I strongly suggest that the authors implement all the suggestions made in these two reviews, to increase the significance of their contribution.